# MARGINAL FLOW: A FLEXIBLE AND EFFICIENT FRAMEWORK FOR DENSITY ESTIMATION

## ABSTRACT

Current density modeling approaches suffer from at least one of the following shortcomings: expensive training, slow inference, approximate likelihood, mode collapse or architectural constraints like bijective mappings. We propose a simple yet powerful framework that overcomes these limitations altogether. We define our model $q_\theta(\boldsymbol{x})$ through a parametric distribution $q(\boldsymbol{x}|\boldsymbol{w})$ with latent parameters $\boldsymbol{w}$. Instead of directly optimizing the latent variables $\boldsymbol{w}$, our idea is to marginalize them out by sampling them from a learnable distribution $q_\theta(\boldsymbol{w})$, hence the name Marginal Flow. In order to evaluate the learned density $q_\theta(\boldsymbol{x})$ or to sample from it, we only need to draw samples from $q_\theta(\boldsymbol{w})$, which makes both operations efficient. The proposed model allows for exact density evaluation and is orders of magnitude faster than competing models both at training and inference. Furthermore, Marginal Flow is a flexible framework: it does not impose any restrictions on the neural network architecture, it enables learning distributions on lower-dimensional manifolds (either known or to be learned), it can be trained efficiently with any objective (e.g. forward and reverse KL divergence), and it easily handles multi-modal targets. We evaluate Marginal Flow extensively on various tasks including synthetic datasets, simulation-based inference, distributions on positive definite matrices and manifold learning in latent spaces of images.

## 1 INTRODUCTION

Density estimation models are ubiquitous in machine learning and have been used for a wide range of purposes. Their overarching characteristic is to provide an approximation to some probability distribution. The most popular use case is probabilistic modeling of data with the goal of generating new instances. The underlying assumption is that there exists an unknown generative process that generated the data in the first place. Successful applications include generation of images, e.g. Rombach et al. (2022), text-to-audio, e.g. Liu et al. (2023), and text-to-video, e.g. Singer et al. (2023). Other popular applications of deep generative models include protein structure prediction, e.g. Abramson et al. (2024), and drug discovery, e.g. Zeng et al. (2022).

Rather than focusing on generating new samples, another interesting use case of density estimation models lies in modeling and reasoning about the probability distribution itself, which has relevant applications in the sciences. Common settings include computation of high-dimensional integrals and intractable likelihoods or posteriors. This is maybe best exemplified by Bayesian inference, e.g. Rezende & Mohamed (2015). Applications include cosmology, e.g. Alsing et al. (2018), neurosciences, e.g. Goncalves et al. (2020), simulation-based inference, e.g. Cranmer et al. (2020), and many more. Learning probability distributions on manifolds is also a challenging problem that can be addressed with density estimation models, e.g. Gemici et al. (2016); Chen & Lipman (2024).

The two fundamental operations that characterize a density estimation model are sampling from the learned distribution and evaluating its probability density. Most models show a trade-off in efficiency between the two operations, which have their own specific challenges. On the one hand, evaluating the probability density often requires restricting the learned transformations to bijections that are carefully designed to avoid computing expensive Jacobian determinants, as in the case of Normalizing Flows (NF) (Kobyzev et al., 2020). Alternatively, the true density can be bounded like in VAEs (Kingma & Welling, 2014; Rezende et al., 2014) and afterwards estimated (Burda et al., 2015), which is still very expensive. Therefore, most generative models rely on surrogate objectives

Table 1: Comparison of Marginal Flow with other deep generative models: GANs, VAEs, Energy-Based models (EB), Flow Matching (FM), Normalizing Flow (NF), and Free-form Flows (FFF). The Table is inspired by Bond-Taylor et al. (2021).

| Feature | GANs | VAEs | EB | FM | NF | FFF | **Ours** |
|---|---|---|---|---|---|---|---|
| Efficient exact likelihood | ✗ | ✗ | ✗ | ✗ | ✓ | ✗ | ✓ |
| Efficient (single-step) sampling | ✓ | ✓ | ✗ | ✗ | ✓ | ✓ | ✓ |
| Efficient training | ✗ | ✓ | (✓) | (✓) | ✗ | (✓) | ✓ |
| Free-form Jacobian | ✓ | ✗ | ✓ | ✓ | ✗ | ✓ | ✓ |
| Lower dim. base distr. (manifold) | ✓ | ✓ | ✗ | ✗ | ✗ | ✓ | ✓ |

that do not require the evaluation of the probability densities, while still allowing for high-fidelity sample generation. This is the case for Energy-Based (EB) models (Swersky et al., 2011), Diffusion models (Sohl-Dickstein et al., 2015) and Flow Matching (FM) (Lipman et al., 2023). On the other hand, sampling often requires multi-step processes that transform samples from a simple distribution into samples from the learned distribution, e.g. Flow Matching and Diffusion models. The trade-off between efficient log-likelihood evaluation and efficient sampling is clear in NF, which can be efficient only at either sampling or evaluating the density. Which of the two operations is more efficient also determines which objective function can be used for training.

In many applications it is beneficial to learn a density on a lower-dimensional space. For instance, real data is often assumed to live on a lower-dimensional manifold (Fefferman et al., 2016). Most models, like Diffusion, FM and NF, cannot account for a change in the dimensionality while others like GANs (Goodfellow et al., 2014) or Free-form Flows (Draxler et al., 2024) can, but suffer from other disadvantages like approximate likelihood and unstable training.

**Contribution.** We propose a novel density estimation framework that alleviates altogether the common shortcomings of current approaches. We define our model through a parametric distribution $q(\boldsymbol{x}|\boldsymbol{w})$ with latent parameters $\boldsymbol{w}$. Instead of directly optimizing the latent variables $\boldsymbol{w}$, we marginalize them out by sampling $\boldsymbol{w}$ from a learnable distribution $q_\theta(\boldsymbol{w})$. As we do not need to evaluate $q_\theta(\boldsymbol{w})$ at any point, but only to sample from it, we are free to generate samples in a very flexible and efficient way. To generate $\boldsymbol{w}$, we feed-forward samples from a base distribution of choice through an unconstrained learnable neural network. Overall, the proposed approach allows for efficient exact density evaluation and efficient sampling. Furthermore, it does not pose any restrictions (e.g. bijectivity) on the neural network and allows for learning a lower-dimensional manifold alongside the density. In Table 1, we provide a high-level comparison between popular density estimation models and Marginal Flow. Overall, our contributions can be summarized as follows:

- We introduce a novel density estimation framework called Marginal Flow.
- We demonstrate the flexibility of the framework: it allows for learning lower-dimensional manifolds, it can easily handle multi-modal distributions, and can be tailored to the data with the choice of the parametric distribution $q(\boldsymbol{x}|\boldsymbol{w})$.
- We show empirically that Marginal Flow is orders of magnitude faster than competing models both at training and inference.
- Lastly, we showcase Marginal Flow on extensive experiments with synthetic data (trained via log-likelihood and reverse KL divergence), simulation-based inference, distributions over positive-definite matrices, and finally on MNIST digits and the JAFFE faces dataset.

## 2 MARGINAL FLOW

### 2.1 MODEL DEFINITION

**Marginalization** Let $q(\boldsymbol{x}|\boldsymbol{w})$ with $\boldsymbol{x} \in \mathbb{R}^d$ be a family of distributions parametrized by $\boldsymbol{w} \in \mathbb{R}^p$ and assume that, for given $\boldsymbol{w}$, it is easy to evaluate the density of $q(\boldsymbol{x}|\boldsymbol{w})$ to sample from it. We can compute $q(\boldsymbol{x})$ by marginalizing out $\boldsymbol{w}$ over some $q(\boldsymbol{w})$:

$$q(\boldsymbol{x}) = \int q(\boldsymbol{x}|\boldsymbol{w})q(\boldsymbol{w})d\boldsymbol{w} = \mathbb{E}_{\boldsymbol{w}\sim q(\boldsymbol{w})}\left[q(\boldsymbol{x}|\boldsymbol{w})\right] . \tag{1}$$

In our model, we let $q(\boldsymbol{x}|\boldsymbol{w})$ be a distribution of choice parametrized by $\boldsymbol{w}$ and we let $q(\boldsymbol{w})$ be freely learnable: $q(\boldsymbol{w}) \to q_\theta(\boldsymbol{w})$. The resulting marginal $q(\boldsymbol{x})$ is universal for many families of distributions $q(\boldsymbol{x}|\boldsymbol{w})$, e.g. if $q(\boldsymbol{x}|\boldsymbol{w})$ is a kernel (Micchelli et al., 2006). We will often assume $q(\boldsymbol{x}|\boldsymbol{w}) = \mathcal{N}(\boldsymbol{x}|\mu = \boldsymbol{w}, \Sigma = \mathrm{diag}(\sigma_1, \ldots, \sigma_d))$, for which $p = d$, and learnable variances (alongside $\theta$). However, we show that other choices of $q(\boldsymbol{x}|\boldsymbol{w})$ can be beneficial, depending on the setting.

**Definition.** Motivated by the marginalization in Eq. 1, we define our model as follows:

$$q_\theta(\boldsymbol{x}) := \frac{1}{N_c} \sum_{i=1}^{N_c} q(\boldsymbol{x}|\boldsymbol{w}_{\theta,i}) \qquad \text{where} \qquad \boldsymbol{w}_{\theta,i} \sim q_\theta(\boldsymbol{w}) \,. \tag{2}$$

The density $q_\theta(\boldsymbol{x})$ can be exactly evaluated and efficiently sampled from. $N_c$ is the number of parameters drawn from $q_\theta(\boldsymbol{w})$ and is not required to be fixed. In fact, the parameters $\boldsymbol{w}_{\theta,i}$ are not fixed themselves but rather *resampled* from $q_\theta(\boldsymbol{w})$ at each iteration, which effectively renders the marginalization in Eq. 1. As we will argue in the next paragraph, there is a crucial difference with respect to directly optimizing a finite set of mixtures $\{\boldsymbol{w}_i\}_{i=1}^{N_c}$. Another important aspect is that we do not need to evaluate $q_\theta(\boldsymbol{w})$ but only to sample from it. Therefore, we can construct samples in a very flexible way and in a single step: we first sample from a distribution of choice $p_{\mathrm{base}}(\boldsymbol{z})$ with $\boldsymbol{z} \in \mathbb{R}^m$ and then transform them via a learnable mapping to the space of latent parameters $\boldsymbol{w} \in \mathbb{R}^p$. Relevantly, to do so we can use an unconstrained learnable function $f_\theta : \boldsymbol{z} \in \mathbb{R}^m \mapsto \boldsymbol{w} \in \mathbb{R}^p$:

$$\boldsymbol{w}_{\theta,i} := f_\theta(\boldsymbol{z}_i) \qquad \text{with} \qquad \boldsymbol{z}_i \sim p_{\mathrm{base}}(\boldsymbol{z}) \,. \tag{3}$$

The resulting samples $\boldsymbol{w}_{\theta,i} := f_\theta(\boldsymbol{z}_i)$ will be samples from some (learnable) distribution $q_\theta(\boldsymbol{w})$. The neural network $f_\theta(\boldsymbol{z})$ is thus the trainable part of the model. In our experiments, a small MLP with 3-5 layers and 256 neurons was enough. Unlike most density estimation models, Marginal Flow is efficient both at sampling and at evaluating the probability density, as we will see in Section 2.2. Furthermore, in contrast to competing models, we can learn a density with support on a lower-dimensional manifold by simply choosing a base distribution with support in $\mathbb{R}^m$ with $m < d$.

**Motivation for marginalization.** In order to understand the importance of the marginalization aspect, consider the case where we have a finite number of $\boldsymbol{w}_i$ and, instead of integrating them out, we optimize them. Without marginalization, the model reduces to a simple mixture model optimized over a fixed set of mixture components $\{\boldsymbol{w}_i\}_{i=1}^{N_c}$, e.g. a Gaussian Mixture Model (GMM) if $q(\boldsymbol{x}|\boldsymbol{w}) = \mathcal{N}(\boldsymbol{x}|\mu = \boldsymbol{w}, \Sigma = \sigma \mathbb{1})$. In this case, learning a target distribution amounts to placing the $N_c$ Gaussians in an optimal way. The expressiveness and scalability of the model are then fundamentally limited by the number of mixtures $N_c$. Instead of optimizing over fixed $\{\boldsymbol{w}_i\}_{i=1}^{N_c}$, our approach relies on the marginalization of $\boldsymbol{w}$, sampled from $q_\theta(\boldsymbol{w})$. We optimize the parameters $\theta$ of the neural network $f_\theta(\boldsymbol{z})$, and we resample $\boldsymbol{w} \sim q_\theta(\boldsymbol{w})$ at each iteration. *The resampling induces an approximation to the marginal distribution in Eq. 1, rather than just a finite mixture.* As illustrated in Figure 1, even with the same nominal number of mixtures (e.g. 10), only the marginalized model is able to learn a smooth density. As such, the modeling capacity is not directly linked to $N_c$ anymore. The marginalization prevents the collapse to a GMM and spreads $q_\theta(\boldsymbol{w})$ to cover the entire target.

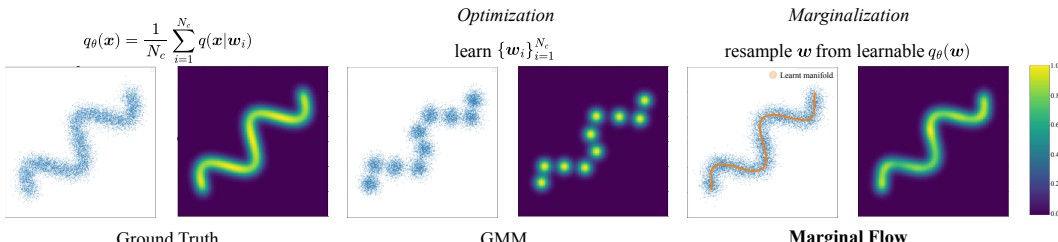

Figure 1: Motivation for marginalization: learned distribution and samples when optimizing directly the parameters $\boldsymbol{w}_i$ compared to resampling them from a learnable $q_\theta(\boldsymbol{w})$, as in Marginal Flow.

## 2.2 EFFICIENT EVALUATION AND SAMPLING

**Sampling the parameters $\boldsymbol{w}_i$ Figure 2 (*left*).** In order to evaluate the modeled density $q_\theta(\boldsymbol{x})$ or to sample from it, we first need to sample $\boldsymbol{w}_i$, which parametrize $q(\boldsymbol{x}|\boldsymbol{w}_i)$. This is done ef-

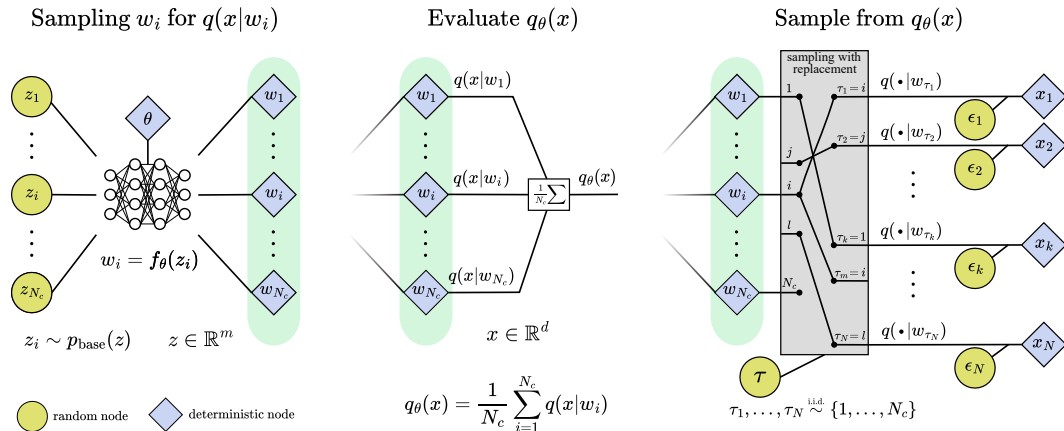

Figure 2: Marginal flow model diagram. Evaluating the modeled density $q_\theta(\boldsymbol{x})$ (*center*) and sampling from $q_\theta(\boldsymbol{x})$ (*right*) requires to first sample the parameters $\boldsymbol{w}_i$ (*left*).

ficiently by feed-forwarding samples $\{\boldsymbol{z}_i\}_{i=1}^{N_c}$ from a base distribution of choice: $\boldsymbol{w}_i = f_\theta(\boldsymbol{z}_i)$ with $\boldsymbol{z}_i \sim p_{\text{base}}(\boldsymbol{z})$. With the sampled $\{\boldsymbol{w}_i\}_{i=1}^{N_c}$, our model in Eq. 2 resembles a mixture model with $N_c$ components. Note, however, that the $\{\boldsymbol{w}_i\}_{i=1}^{N_c}$ *are not fixed but sampled again for each evaluation or sampling of* $q_\theta(\boldsymbol{x})$. The neural network $f_\theta$ is unconstrained. **Evaluation: Figure 2 (*center*).** In order to evaluate the density $q_\theta(\boldsymbol{x})$ at a given point $\boldsymbol{x}$, we use the definition in Eq. 2. Given the sampled parameters $\{\boldsymbol{w}_i\}_{i=1}^{N_c}$, we only need to evaluate each $q(\boldsymbol{x}|\boldsymbol{w}_i)$ on $\boldsymbol{x}$, which is chosen to have a simple closed-form density function. Note that, in contrast to other density estimation models, the evaluation of the density does not require inverting $f_\theta(\boldsymbol{z}_i)$, computing $\det \mathcal{J}_{f_\theta}$ or solving an ODE. **Sampling from $q_\theta(\boldsymbol{x})$: Figure 2 (*right*).** Sampling as in Eq. 2 is also efficient, just like sampling from a mixture model. Given the sampled parameters $\{\boldsymbol{w}_i\}_{i=1}^{N_c}$, we first need to sample a component $\boldsymbol{w}_j$ and then sample from the associated distribution $q(\boldsymbol{x}|\boldsymbol{w}_j)$, with $j \in \{1, \ldots, N_c\}$. To draw $N$ samples, we sample $N$ indices with replacement from $\{1, \ldots, N_c\}$.

**Empirical runtime.** We now empirically measure runtime for sampling and evaluating the exact density and compare against competing models. Note that only Marginal Flow and Normalizing Flow (NF) provide exact density by construction. As shown in Figure 3, Marginal flow is orders of magnitude faster than competing methods in terms of both sampling and density evaluation, where FM is Flow Matching and FFF is Free-form Flows. Sampling is as efficient as in FFF, since both only require drawing from a base distribution and passing the samples through a neural network. For further details, see the Appendix in Section A.3.1.

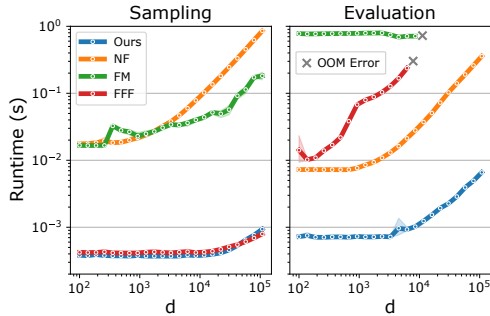

Figure 3: Runtime for sampling (*left*) and exact density evaluation (*right*) of 100 points.

### 2.3 FLEXIBILITY OF MARGINAL FLOW

**Lower-dimensional latent distribution.** Most density estimation models, like Flow Matching and Normalizing Flows, learn mappings that preserve the dimensionality and cannot learn densities on lower-dimensional manifolds. Some work tries to overcome this issue either by resorting on approximations (Brehmer & Cranmer, 2020) or by restricting the transformations (Khorashadizadeh et al., 2023; Negri et al., 2025). In contrast, with our model in Eq. 2, we have the freedom of choosing the dimensionality of the base distribution, i.e. $p_{\text{base}}(\boldsymbol{z})$ with support in $\mathbb{R}^m$ with $m < d$. Also in this case we can evaluate $q_\theta(\boldsymbol{x})$ exactly and learn the manifold alongside the density. In Figure 4 we showcase Marginal Flow and competing models on a density defined on a (unknown) 1D manifold.

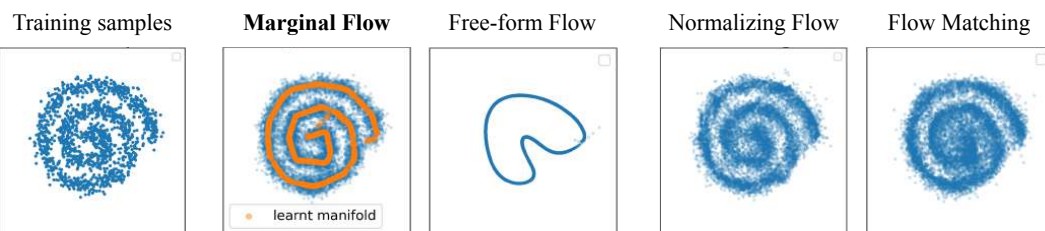

Figure 4: Toy example of density defined on (unknown) 1D manifold. (*Left*) Training data consists of 1500 points. (*center*) Marginal Flow perfectly learns the density and discovers the correct manifold. Free-form Flow learns an incorrect manifold and is not able to embed the density in 2D space. (*right*) Flow Matching and Normalizing Flow learn the density but cannot account for a manifold.

**Conditional distribution.** As wo do not have any requirements on the neural network $f_\theta(z)$, Marginal Flow can be readily extended to model conditional distributions. The conditioning variables could be appended to the input $f_\theta(z) \to f_\theta(z; c)$ or one could use a hypernetwork that takes $c$ as input and returns the neural network parameters $f_\theta(z) \to f_{\theta(c)}(z)$. Furthermore, the base distribution can also be conditioned on $c$: $p_{\text{base}}(z) \to p_{\text{base}}(z; c)$.

**Multi-modal targets.** Marginal Flow can naturally account for multi-modal targets thanks to the unconstrained neural network $f_\theta(z)$. Most generative models, like Normalizing Flows and Flow Matching, learn (directly or indirectly) a bijection between a base distribution and the target distribution. However, bijections struggle to learn new modalities and have limited expressiveness (Liao & He, 2021). Even with a multi-model base distribution, bijections will still struggle to match the modalities in the target with those of the base distribution. Furthermore, many density estimation models suffer from mode collapse during training (He et al., 2019; Kossale et al., 2022). In Figure 5 we showcase how easily Marginal Flow can learn multi-modal targets compared to other models.

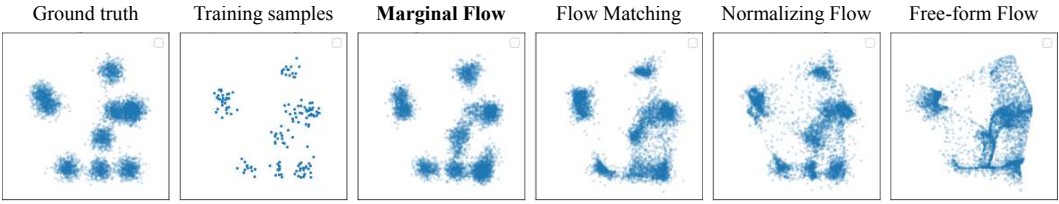

Figure 5: Toy example of multi-modal density learned by log-likelihood on 150 data points. For a fair comparison, all models use a uniform base distribution. Note that Marginal Flow is not a mixture model (for which this task would be trivial) since $w_i$ are always resampled (see Figure 1).

**Training objectives.** Density estimation models are usually trained through an objective that requires sampling, evaluating the (exact) density or both. However, current approaches are efficient only at either one or the other. For instance, models trained on data via forward KL divergence (i.e. log-likelihood) require efficient density evaluation while models trained on unnormalized targets via reverse KL divergence require efficient sampling. However, one could wish to use both objectives to combine information from observations and unnormalized targets or to mitigate the mean-seeking (mode-seeking) behavior of the forward (reverse) KL divergence. Since Marginal Flow is efficient both at sampling and evaluation, it can be trained efficiently with most objectives; see Appendix A.2.

**Extension to other mixtures.** The proposed model in Eq. 2 leaves complete freedom in the choice of $q(x|w)$, as long as it can be parametrized by some $w$. In most experiments we employ a Gaussian with learnable variances, i.e. $q(x|w) = \mathcal{N}(x|\mu = w, \Sigma = \text{diag}(\sigma_1, \dots, \sigma_d))$. However, other choices are possible depending on the application. For instance, when modeling distributions on the probabilistic simplex, we can use the Dirichlet distribution. We can model distributions on symmet-

ric positive-definite matrices by choosing $q(\boldsymbol{x}|\boldsymbol{w})$ to be a Wishart, which we showcase in Section 4.3. Relevantly, the choice of $q(\boldsymbol{x}|\boldsymbol{w})$ does not affect the structure of the proposed framework.

## 3 RELATED WORK

One of the earliest attempts to use deep learning for generative modeling are Energy-based (EB) models (Swersky et al., 2011). Instead of modeling a normalized density, EB models learn the negative log-probability. Despite their flexibility, computing the exact density and sampling from the model is generally expensive (Song & Kingma, 2021). Closely related are diffusion models (Sohl-Dickstein et al., 2015), which learn how to reverse a fixed noising process by estimating at each step the gradient of the log-density. Diffusion models can produce high-quality samples (Rombach et al., 2022; Liu et al., 2023), but still require multi-step sampling and do not provide the exact density.

Another approach is to model the observed density with unobserved latent variables. VAEs (Kingma & Welling, 2014; Rezende et al., 2014) encode data into a latent space and are trained via a lower bound on the log-likelihood. In contrast to EB models, VAEs can be sampled in a single step. However, VAEs have limited expressiveness and suffer from posterior collapse (He et al., 2019). Another latent variable model – GANs (Goodfellow et al., 2014) – consists of a generator that creates samples from a latent distribution and a discriminator trained to distinguish generated samples from real ones. GANs can generate high-fidelity images (Karras et al., 2019) but are unstable and suffer from mode collapse (Kossale et al., 2022). Neither GANs nor VAEs provide the exact likelihood.

Normalizing Flows (NFs) (Papamakarios et al., 2021) provide a principled way to compute the exact density. NFs transform a base distribution through bijections and account for the probability change via the Jacobian determinant, which is expensive to compute. Thanks to their exact density, NF have been applied for posterior approximations (Rezende & Mohamed, 2015). Additional limitations of NFs arise from the limited expressivity of bijective layers (Liao & He, 2021). Efficiency could be obtained using approximate bijections and by approximating the Jacobian determinant (Draxler et al., 2024), which however precludes sound statistical understanding and evaluation of the exact log-likelihood. Lipman et al. (2023) proposed to learn instead a velocity field that transforms the base distribution into the target. While this approach scales to high-dimensions, it cannot handle lower-dimensional base distributions and still requires expensive ODE solvers to compute the exact density. For a comprehensive review on generative models we refer to Bond-Taylor et al. (2021).

## 4 EXPERIMENTS

First, we show on synthetic data that Marginal Flow can learn complex distributions both via log-likelihood and reverse KL divergence training. We also show that it converges more quickly than competing models. Second, we showcase how Marginal Flow can learn complex conditional distributions and achieve state-of-the-art results for simulation-based inference. Third, we show that Marginal Flow can be easily adapted to learn distributions on positive-definite matrices by simply changing the parametric form of $q(\boldsymbol{x}|\boldsymbol{w})$. Lastly, we showcase applications in computer vision as well: we learn densities on lower-dimensional manifolds on MNIST and on the JAFFE face dataset.

### 4.1 SYNTHETIC DATASETS

**Log-likelihood training.** As illustrative examples, we picked 4 common synthetic datasets (*Two moons*, *Pinwheel*, *Swiss Roll* and *Checkerboard*) and 1 additional multi-modal distribution (*Mixture of Gaussians*). We train Marginal Flow by maximizing the log-likelihood, which is reported explicitly in the Appendix 6. In Figure 6 we showcase that Marginal Flow can perfectly learn all densities without needing any fine-tuning. Next, we study the ability of Marginal Flow to learn densities when a limited number of observations is available. In particular, we compare against Flow Matching, Normalizing Flow and Free-form Flows with an increasing number of training points $\{100, 200, 500, 1000\}$. For a fair comparison we used a comparable amount of parameters in each model. In the Appendix in Figure 13, we show the learned densities, which are particularly accurate for Marginal Flow, already in few-sample regimes. In Figure 7 we showcase the test log-likelihood during training for all models and datasets when train on 1000 points. Marginal Flow converges orders of magnitude quicker than competing models.

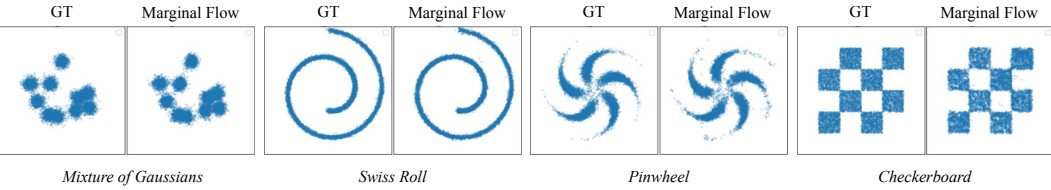

Figure 6: Marginal Flow trained via log-likelihood on 2D synthetic datasets. We show 10'000 samples from the true distribution and from Marginal Flow.

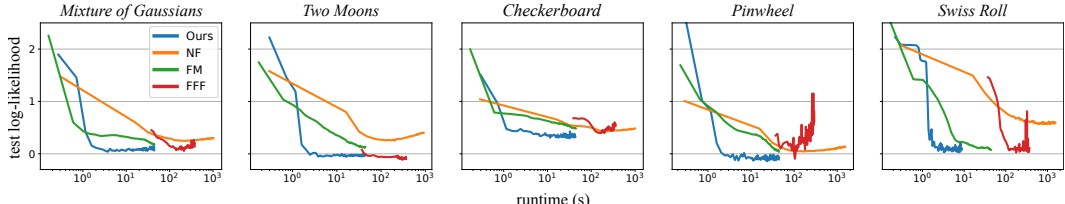

Figure 7: Test log-likelihood of Marginal Flow and other models during training with 1000 points.

**Reverse KL divergence training** We additionally show that Marginal Flow can be trained in the reverse KL direction as well, namely without observations and only guided by the (unnormalized) density of the target distribution. This type of training requires an efficient computation of the exact log-likelihood, which is possible only for Normalizing Flow. Some attempts to make Flow Matching work in this direction have been made but remain limited (Tong et al., 2024). We tried with a score-matching objective but it led to unstable training. We trained Marginal Flow and Normalizing Flow with a reverse KL objective and compared the learned densities in terms of test KL. Marginal Flow achieved superior or comparable performance with Normalizing Flow, see Figure 8 (*left*), and showed better density reconstruction quality, see Figure 8 (*right*). Note that we do not use the *Checkerboard* dataset because its density is constant and has gradients equal to zero everywhere.

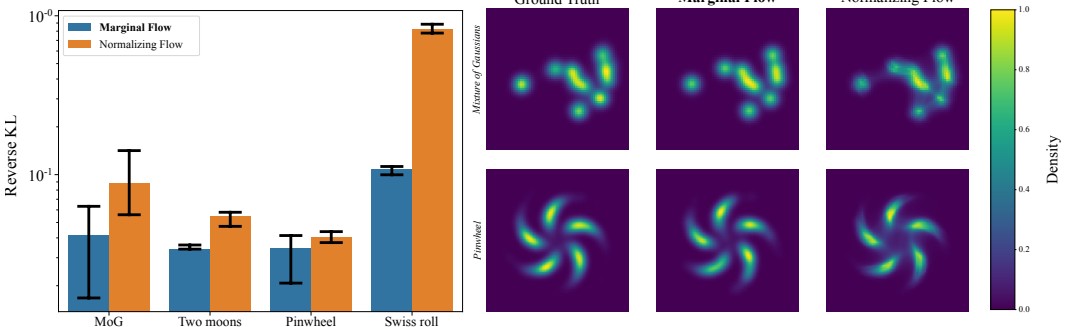

Figure 8: Marginal Flow vs Normalizing Flows trained by reverse KL divergence on synthetic distributions. During training only the probability density is queried (no observations). (*left*) Test reverse KL with 95% confidence intervals error bars. (*right*) comparison of learned density distributions.

## 4.2 SIMULATION-BASED INFERENCE

As argued in Section 2.3, with the proposed framework we can easily learn conditional distributions as well. We showcase Marginal Flow on complex conditional distributions by training it on the Simulation-Based Inference (SBI) benchmark (Lueckmann et al., 2021). SBI data consists of tuples $\{\boldsymbol{x}_i, \theta_i\}_i$, where $\theta_i$ are parameters sampled from a prior $p(\theta)$ and $\boldsymbol{x}_i$ are samples from a simulator $p(\boldsymbol{x}|\theta_i)$ parameterized by $\theta_i$. Given tuples of observations $\{\boldsymbol{x}_i, \theta_i\}_i$, the goal is to learn the posterior $p(\theta|\boldsymbol{x}_j)$ of a new $\boldsymbol{x}_j$. Evaluation is performed in terms of Classifier 2-Sample Tests (C2ST) on a held-out test set. Due to space constraints we report results in the Appendix in Figure 14. Marginal Flow achieves state-of-the-art results and proves to be particularly effective in low data regimes.

### 4.3 WISHART MIXTURE DISTRIBUTION

One interesting aspect of Marginal Flow is that the parametric family $q(\boldsymbol{x}|\boldsymbol{w})$ in Eq. 2 can be adjusted depending on the application and on the noise assumption. Consider the case of learning a Wishart mixture distributions (Haff et al., 2011; Cappozzo & Casa, 2025): observations consist of sample covariances, which lie on the cone of positive-definite (p.d.) matrices. One design choice would be to use a Gaussian assumption in $q(\boldsymbol{x}|\boldsymbol{w})$ and then transform the samples into positive definite matrices through bijective layers as in Negri et al. (2023). Alternatively, one could directly choose $q(\boldsymbol{x}|\boldsymbol{w})$ to be Wishart distributions. We showcase this second option, and, in particular, we parametrize the scale matrices of Wishart via $\boldsymbol{w}_i$, in addition to a parametrized global $\nu$. We consider a target distribution $t(\boldsymbol{x})$ where the generating parameters live on a 1D manifold:

$$t(\boldsymbol{x}) = \mathcal{W}(\boldsymbol{x}; \nu, \Sigma(\lambda)) \quad \text{s.t.} \quad \Sigma(\lambda) \in \mathcal{M} \quad \forall \lambda \in [0, 1] . \tag{4}$$

We showcase training using both the reverse and forward KL divergence (log-likelihood). Our goal is to approximate $t(\boldsymbol{x})$ while reconstructing the manifold $\mathcal{M}$. We showcase two settings. (i) A low-dimensional setting with $10 \times 10$ matrices using the reverse KL and we compare to Normalizing Flows (NFs) parameterizing the Cholesky factor. (ii) A high-dimensional setting with $100 \times 100$ matrices using the forward KL, which was computationally prohibitive for NFs. In Figure 9 we show test KL divergence in the low-dim setting and plot the manifold reconstruction using a PCA projection to 2D. Marginal Flow perfectly recovers the manifold in both training directions and approximates $t(\boldsymbol{x})$ better than NFs. For more details on the target manifold $\mathcal{M}$ see Appendix A.4.2.

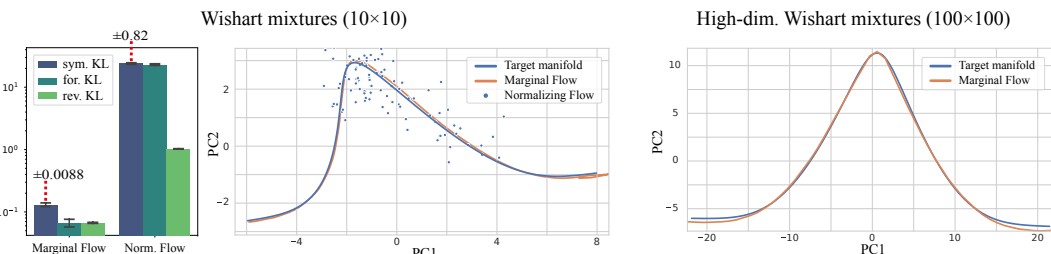

Figure 9: (*left*) $10 \times 10$ Wishart mixtures ($d = 55$) on manifold trained via reverse KL. Test KL divergences in the bar plot show accurate fit with Marginal Flow and underfitting with NF. Unlike NF, we can also learn the manifold. (*right*) Reconstructed manifold for $100 \times 100$ Wishart mixtures ($d = 5050$) trained via forward KL (log-likelihood). NF cannot be trained in such a high-dim setting.

### 4.4 MANIFOLDS IN IMAGE LATENT-SPACES

Most modern image generative models rely on non-trivial latent spaces, e.g. Rombach et al. (2022), which can still be relatively high-dimensional and show non-Euclidean behavior (Shao et al., 2018). It would then be relevant to traverse such latent spaces on a lower-dimensional manifold. Marginal Flow is well-suited for this task since it allows for learning a lower-dimensional manifold alongside the density. We showcase this on MNIST digits (LeCun et al., 1998) and the JAFFE face dataset (Lyons et al., 1998). The JAFFE dataset contains 214 face images of ten Japanese women mimicking certain emotions. Each image is associated with a score quantifying the emotions, e.g. "happiness" or "surprise". Note that learning a manifold with such little data is very challenging.

In both settings, we first train a VAE without conditional information to encode images into a latent space (20- and 10-dimensional, respectively). Then, we train a single Marginal Flow in the latent space to learn a low-dimensional manifold conditioned on the digit label (or emotion score). The exact loss function is reported in the Appendix in Eq. 8. In particular, we use a 1-dim uniform base distribution $p_{\text{base}} = \mathcal{U}([-1, 1])$. We learn conditional manifolds via the network $f_\theta(\boldsymbol{z}; c)$, with $\boldsymbol{z} \in [-1, 1]$ and $c$ the class label (or scores). In Figure 10, we explore the 1-dim manifold conditioned on each label of **MNIST**. Results show similarities across digits in the learned manifold: some sections look approximately **bold**, ***bold italic*** and normal font, with smooth transitions in between them. For **JAFFE**, the manifold smoothly interpolates the different faces (horizontally) at fixed emotion levels, as shown in Figure 11. We observe disentanglement of faces and emotions, as faces tend to align within columns. Some inconsistencies are probably the result of the extremely low-data regime. For further visualizations, see the Appendix, Figure 15 and 16.

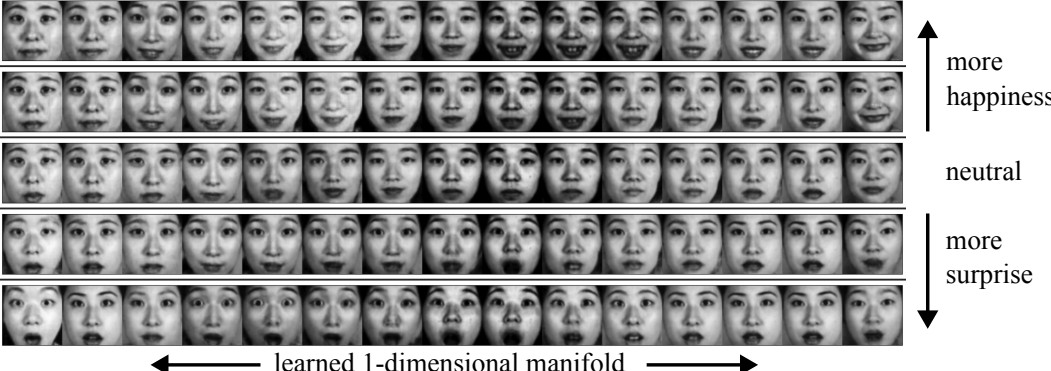

Figure 10: Each row shows the 1-dim manifold conditioned on the label learned by Marginal Flow on MNIST (in a 20-dim VAE latent space). We observe disentanglement of digits and writing style.

Figure 11: By traversing the conditional manifold, Marginal Flow smoothly interpolates between faces and levels of emotions on the JAFFE dataset. While the conditioning value is kept fixed in each row, columns correspond to the same point on the learned manifold in the latent space.

## 5 CONCLUSIONS

In this work we introduced a flexible and efficient density estimation framework called Marginal Flow. We showed empirically that Marginal Flow is orders of magnitude faster than competing methods in terms of runtime, both at sampling and exact density evaluation. Unlike most density estimation models, Marginal Flow provides exact density evaluation by construction. Marginal Flow is also a very flexible framework: it allows for learning lower-dimensional manifolds, it can easily handle multi-modal distributions, and it can be easily tailored to the data with the choice of the parametrized distribution $q(x|w)$. Experimentally, we showcase Marginal Flow on several datasets and various tasks. First, we showed that Marginal Flow can perfectly reconstruct synthetic datasets both when trained via log-likelihood and via reverse KL divergence. Additionally, Marginal Flow converges orders of magnitude faster than competing models. Then, we showed that it can achieve state-of-the-art results on the Simulation-based Inference benchmark. We also showed that we can easily adapt Marginal Flow to learn distributions on positive definite matrices by choosing the Wishart distribution as the parametrized family $q(x|w)$. Lastly, we applied Marginal Flow to learn a (conditional) manifold alongside the density for MNIST digits and the JAFFE face dataset.

## REPRODUCIBILITY

We made an effort to make every aspect of the model and of the experiments reproducible. In particular, as part of the submission we provide code with a `PyTorch` implementation of the model and code for reproducing figures and experiments. Furthermore, in Appendix A.1 we discuss implementation details of Marginal Flow concerning sampling, log density evaluation and neural network architecture. Finally, in Appendix A.3 we provide detailed description of the experiments conducted including data pre-processing for real-world experiments.

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

# A  APPENDIX

## A.1  IMPLEMENTATION DETAILS

We provide our implementation of Marginal Flow in `PyTorch` (Paszke et al., 2019) as part of the supplementary material. Here we discuss the main high-level aspects of such an implementation.

**Density evaluation and sampling.**  Once we sample the parameters $w$, Marginal Flow consists of a Mixture of distributions $q(x|w)$ parameterized by the sampled $w$. The parameters $w$ are then resampled each time we evaluate the density $q_\theta(x)$ or sample from $q_\theta(x)$, with $q_\theta(x)$ being defined in Eq. 2. Within `PyTorch` one can define the parametric family $q(x|w)$ by simply choosing a distribution of choice from `torch.distributions`. For all distributions, PyTorch provides efficient evaluation of the density and efficient sampling, which can be automatically extended to mixtures of distributions. In most of our experiments we used a Gaussian family, i.e. $q(x|w) = \mathcal{N}(x|\mu = w, \Sigma = \text{diag}(\sigma_1, \ldots, \sigma_d))$. In such a case, one can evaluate the log-density even more efficiently and does not need to rely on `torch.distributions`. In particular, we need to evaluate $N$ points over a mixture with $N_c$ components. This requires computing the distance of each point to each mixture component and then summing the contributions. With `torch.cdist` this operation can be done extremely efficiently.

**Neural network architecture.**  A key aspect of the proposed Marginal Flow is that it leaves complete freedom in the choice of the neural network architecture. In particular, for all our experiments, it was sufficient to we use very simple MLP architectures with 3 to 5 layers and 128 to 256 hidden units. We also employed skip connections. The specific settings used in each experiment can be found in the code provided in the supplementary. For conditional experiments we used a slight modification of the mentioned MLP structure. In particular, we simply appended the conditioning variable(s) to the input. In order to extract high-frequency signals from the (low-dimensional) conditioning variables, we used Fourier features (Tancik et al., 2020)

## A.2  OBJECTIVE FUNCTIONS

Marginal Flow provides efficient exact density evaluation and efficient sampling. Consequently, it can be trained efficiently using most objective functions. Among the most popular ones are the forward KL divergence (log-likelihood) and the reverse KL divergence. The former is the most commonly used one in deep generative models and is employed to learn the distribution of some given data $\mathcal{D} = \{x_j\}_{j=1}^N$. The latter is most commonly used when only an unnormalized target distribution $t(x)$ is known. Below we report the definitions of both objectives and their analytical expression when Marginal Flow is used, i.e. Eq. 2.

**Forward KL (log-likelihood)**  Assume we are given a dataset of observations $\mathcal{D} = \{x_j\}_{j=1}^N$ and the goal is to estimate the unknown distribution that generated the dataset. The underlying assumption is $x_j \sim p(x)$, with $p(x)$ being unknown. The most common approach is to minimize the forward KL divergence, which is proportional to the negative log-likelihood:

$$\mathcal{L}(\theta) = \text{KL}(p(x)||q_\theta(x)) = \int p(x) \log \frac{p(x)}{q_\theta(x)} dx = -\mathbb{E}_{x \sim p(x)}[\log q_\theta(x)] + \text{const} . \quad (5)$$

Given the data points $\{x_j\}_{j=1}^N$, we can approximate the above expression with the following Monte Carlo estimate:

$$\mathcal{L}(\theta) \approx -\frac{1}{N} \sum_{j=1}^N \log q_\theta(x_j) = -\frac{1}{N} \sum_{j=1}^N \log \frac{1}{N_c} \sum_{i=1}^{N_c} q(x_j|w_i) \quad \text{with} \quad w_i \sim q_\theta(w) . \quad (6)$$

In the last equality we used Marginal Flow as variational family $q_\theta(x)$, i.e. Eq. 2. Recall that $q_\theta(w)$ is not modeled explicitly. Instead, we construct samples $w_i$ by transforming samples from a base distribution $p_{\text{base}}(z)$ with a learnable function $f_\theta : z \in \mathbb{R}^m \mapsto w \in \mathbb{R}^d$:

$$w_i := f_\theta(z_i) \qquad \text{with} \qquad z_i \sim p_{\text{base}}(z) . \quad (7)$$

When using a conditional model, the modeled density depends on the conditioning parameter as well: $q_\theta(x) \to q_\theta(x; c)$. One straightforward way to model conditional density with Marginal

Flow is to condition the neural network on $\boldsymbol{c}$, i.e. $f_\theta(\boldsymbol{z}) \rightarrow f_\theta(\boldsymbol{z}; \boldsymbol{c})$ or, more explicitly, $f_{\theta(\boldsymbol{c})}(\boldsymbol{z})$. Assume we are given pairs of observations and conditioning information $\{\boldsymbol{x}_j, \boldsymbol{c}_j\}_{j=1}^N$. Then, the loss function in Eq. 6 reads as:

$$\mathcal{L}(\theta) \approx -\frac{1}{N} \sum_{j=1}^N \log q_\theta(\boldsymbol{x}_j; \boldsymbol{c}_j) = -\frac{1}{N} \sum_{j=1}^N \log \frac{1}{N_c} \sum_{i=1}^{N_c} q(\boldsymbol{x}_j | \boldsymbol{w}_{\boldsymbol{c}_j, i}) \tag{8}$$

$$\text{where} \quad \boldsymbol{w}_{\boldsymbol{c}_j, i} = f_\theta(\boldsymbol{z}_i; \boldsymbol{c}_j) \quad \text{with} \quad \boldsymbol{z}_i \sim p_{\text{base}}(\boldsymbol{z}) \ .$$

**Reverse KL** In variational inference settings we are commonly given an unnormalized target distribution $t(\boldsymbol{x}) \propto p(\boldsymbol{x})$ and we would like to (i) approximate it and (ii) draw samples from it. This is often the case in Bayesian inference: given a likelihood $p(\mathcal{D}|\Theta)$ and a prior $p(\Theta)$, we would like to perform variational inference on the posterior $p(\Theta|\mathcal{D}) \propto p(\mathcal{D}|\Theta)p(\Theta)$, which we can evaluate only up to a constant. We now detail how to train the proposed model to approximate the target distribution $p(\boldsymbol{x})$, which corresponds to $p(\Theta|\mathcal{D})$ in the previous Bayesian posterior inference example. The most common distance measure in variational inference is the reverse Kullback-Leibler divergence, which is defined as

$$\mathcal{L}(\theta) = \text{KL}(q_\theta(\boldsymbol{x}) || p(\boldsymbol{x})) = \int q_\theta(\boldsymbol{x}) \log \frac{q_\theta(\boldsymbol{x})}{p(\boldsymbol{x})} d\boldsymbol{x} = \mathbb{E}_{\boldsymbol{x} \sim q_\theta(\boldsymbol{x})} \left[ \log \frac{q_\theta(\boldsymbol{x})}{p(\boldsymbol{x})} \right] \ . \tag{9}$$

Usually, we do not have access to the normalized $p(\boldsymbol{x})$ but only to some unnormalized target $t(\boldsymbol{x})$, i.e. $p(\boldsymbol{x}) = t(\boldsymbol{x})/\mathcal{N}$. However, the reverse KL divergences are proportional up to a constant, which is precisely the normalization constant $\mathcal{N}$:

$$\text{KL}(q_\theta(\boldsymbol{x}) || p(\boldsymbol{x})) = \text{KL}(q_\theta(\boldsymbol{x}) || t(\boldsymbol{x})) + \log \mathcal{N} \ . \tag{10}$$

In practice, the reverse KL divergence is approximated in Monte Carlo fashion by drawing $N$ samples from the variational distribution $\{\boldsymbol{x}_j\}_{j=1}^N$ with $x_j \sim q_\theta(\boldsymbol{x})$, which gives the following objective:

$$\mathcal{L}(\theta) \approx \frac{1}{N} \sum_{j=1}^N \log \frac{q_\theta(\boldsymbol{x}_j)}{t(\boldsymbol{x}_j)} = \frac{1}{N} \sum_{j=1}^N \log \frac{\frac{1}{N_c} \sum_{i=1}^{N_c} q(\boldsymbol{x}_j | \boldsymbol{w}_i)}{t(\boldsymbol{x}_j)} \quad \text{with} \quad \boldsymbol{w}_i \sim q_\theta(\boldsymbol{w}) \ . \tag{11}$$

In the last equality we plugged in the proposed model in Eq. 2 as variational family $q_\theta(\boldsymbol{x})$. Note that, as opposed to the forward KL divergence setting (log-likelihood), in the reverse KL setting we need to draw samples from the model $\boldsymbol{x}_j \sim q_\theta(\boldsymbol{x})$.

## A.3 EXPERIMENTAL DETAILS

### A.3.1 RUNTIME COMPARISON

In Figure 3 we have shown a runtime comparison for the two main operations of density estimation models: sampling and evaluation of the log-probability. In particular, we measure the runtime for generating 100 samples and for evaluating the log-probability of 100 points. We repeat this operation 10 times per dimension and report the average and 95% confidence intervals. We compare against competing models: Marginal Flow, Flow Matching. Normalizing Flow and Free-form Flow. Marginal Flow and Normalizing Flow naturally provide access to the exact log-likelihood, while Flow Matching does not require it during training, and Free-form Flow uses an approximation. In both cases computing the exact density is computationally expensive. In order to make a fair comparison, we defined all models to have a similar (and small) number of trainable parameters, around $100k$. In particular, for all models (except Normalizing Flows) we employed a simple MLP with 3 layers and 128 neurons each. For Normalizing Flow, which requires bijections, we use 3 coupling layers with splines. Among the many choices of bijective layers, we chose the most efficient ones in terms of runtime, even though such layers are sometimes unstable during training. We ran all runtime experiments on the same consumer-grade A100 GPU with 40 GB of memory. Results show that Marginal Flow is orders of magnitude faster than competing models. In the common log-likelihood training setting, this is relevant both for training (where one needs to repeatedly evaluate the log density) and for inference (in order to generate new samples). Furthermore, results in Figure 7 suggest that Marginal Flow also has better convergence rates.

## A.4 SYNTHETIC EXPERIMENTS

In order to make the comparison among models fair, we made sure to use a comparable amount of parameters. In particular, in all models except Normalizing Flows we used an MLP with 5 layers and 256 neurons. For Normalizing Flow, we used 5 layers of invertible Resnet (Behrmann et al., 2019), which are more expressive (but more computationally expensive) than coupling layers with splines.

**Forward KL divergence training (log-likelihood).**   In the log-likelihood settings, we trained for 5000 epochs and selected the best model on the validation set. In synthetic datasets we could always use full-batch training. We trained over different numbers of data points, i.e. $\{100, 200, 500, 1000\}$, and set $N_c$ to half of the number of training points in each setting. We did not perform any hyperparameter tuning on Marginal Flow. We report additional results with log-likelihood training in Figure 13.

**Reverse KL divergence training.**   In the reverse KL divergence setting we do not have observations, and we need to sample from the modeled densities. This training setting is only viable for Marginal Flow and Normalizing Flow. In both cases we drew 10'000 samples per iteration. Furthermore, during training we used simulated annealing (Kirkpatrick et al., 1983) to explore the full support of the target distribution. In particular, we introduce an artificial temperature $T_i$ for the target distribution in Eq. 10:

$$p_i^*(\boldsymbol{x}) = p(\boldsymbol{x})^{1/T_i} \, , \tag{12}$$

where $T_i$ is the temperature at the $i$-th training iteration. The temperature $T_i$ is slowly annealed during training from the initial $T_0 = 5$ to $T_N = 1$. Note that $p_i^*(\boldsymbol{x}) = p(\boldsymbol{x})$ for $T_i = 1$, which is the true target. If the initial temperature is high enough, $p_i^*$ will likely be very flat, allowing for a better exploration of the support of the distribution. In order to account for the slow annealing of the temperature, we trained for 10'000 iterations. We report a visualization of the density learned by Marginal Flow and Normalizing Flow for all studied densities in Figure 12. Note that we do not train the models on the *Checkerboard* dataset because the true density is constant everywhere and the gradient is thus zero everywhere.

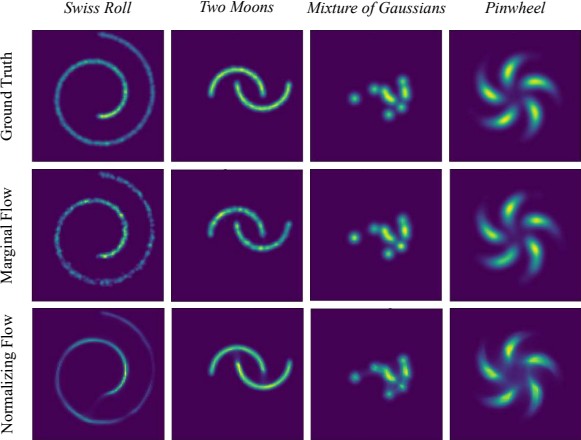

Figure 12: Marginal Flow is trained by reverse KL divergence on 4 synthetic datasets. We evaluate the learned density and compare it with Normalizing Flows.

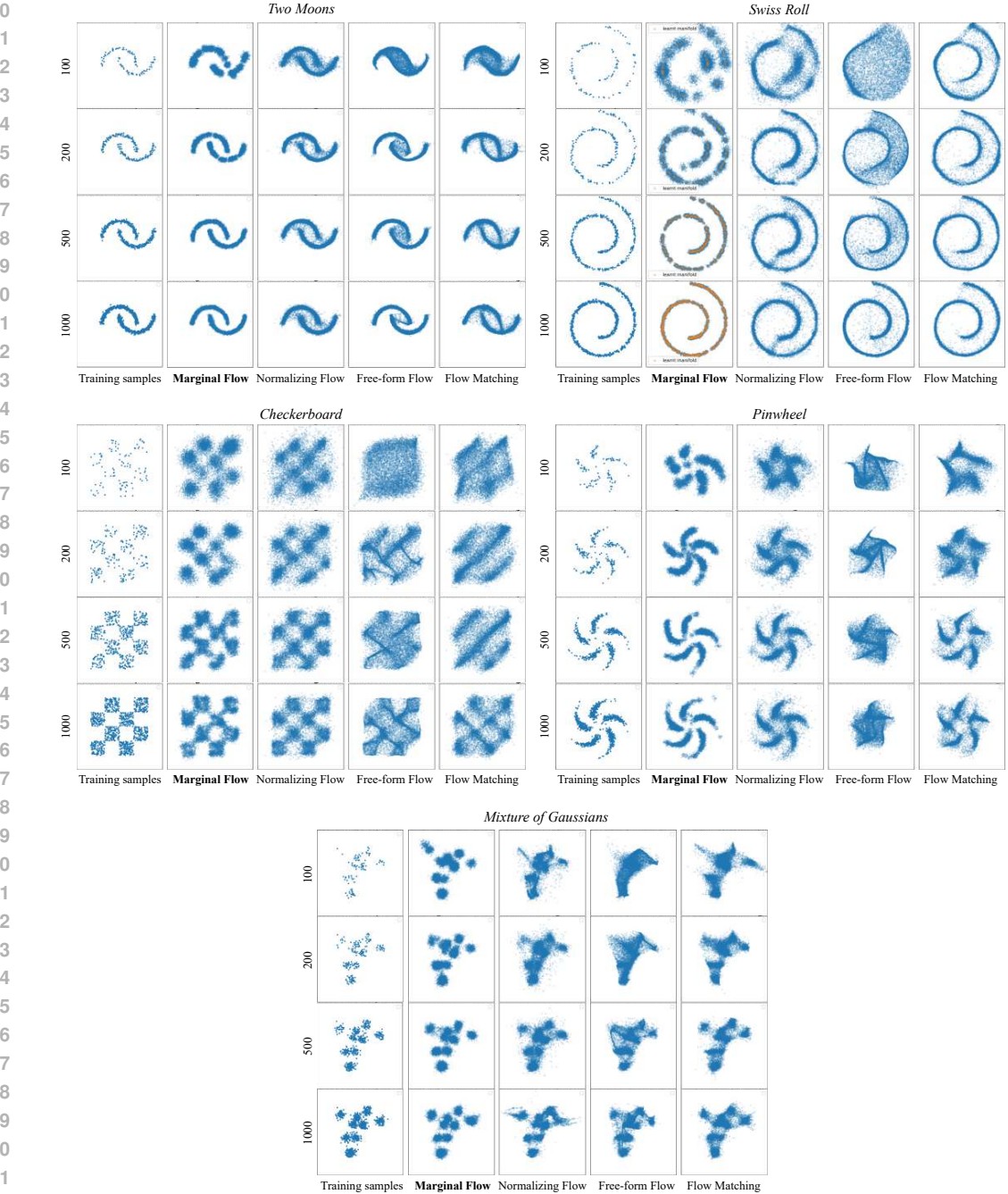

Figure 13: Marginal Flow is trained by forward KL divergence (log-likelihood) on 5 synthetic datasets with increasing number of training points $\{100, 200, 500, 1000\}$. We compare Marginal Flow with Normalizing Flow, Free-form Flow, Flow Matching. Results show that Marginal Flow learns the correct density with fewer samples compared to competing models

### A.4.1 SBI BENCHMARK

In the Simulation-Based Inference benchmark, each setting is provided with three sets of observations with 1000, 10'000, 100'000 points. For each dataset we train Marginal Flow for 2000, 1000 and 250 epochs, respectively. In all cases we trained a Marginal Flow with an MLP with 4 layers and 256 neurons each and $N_c = 2048$. We selected the best model on the validation set and did not perform any other hyperparameter tuning.

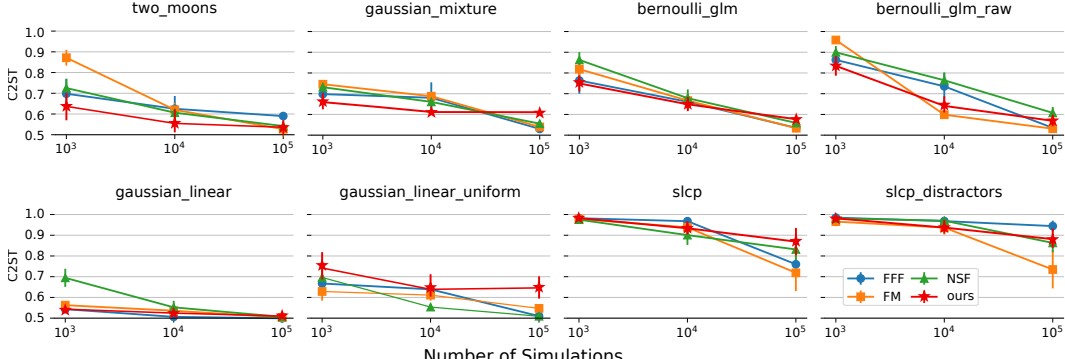

Figure 14: Simulation-based inference benchmark: we show average and standard deviation over 10 different test observations. We compare our method against Free-form Flows (FFF), Flow Matching (FM), and Normalizing Flows (NSF). Benchmark results are taken from Draxler et al. (2024).

### A.4.2 WISHART MIXTURE EXPERIMENT

**Bijection mapping to p.d. matrices.** Marginal Flow and Normalizing Flow employ the same mapping to positive definite (p.d.) matrices. A vector $x$ is reshaped to a lower triangular matrix $L$. Afterwards, the diagonals are transformed to be positive, leading to $L^+$. Finally, if the full covariance matrix is required, then $L^+(L^+)^T$ is computed. The change in Jacobian determinant of each step can be computed efficiently (Negri et al., 2023).

**Target manifold.** We show again the target distribution from Eq. 4 for convenience:

$$t(\boldsymbol{x}) = \mathcal{W}(\boldsymbol{x}; \nu, \Sigma(\lambda)) \quad \text{s.t.} \quad \Sigma(\lambda) \in \mathcal{M} \quad \forall \lambda \in [0, 1] \ . \tag{13}$$

The manifold $\mathcal{M}$ is a straightforward interpolation between covariance matrices with a random structure. Given the covariance matrices $\Sigma_1, \Sigma_2, \Sigma_3 \sim \mathcal{W}(\tilde{\nu}, I)$, the manifold is defined as:

$$\mathcal{M} = \left\{ \Sigma(\lambda) \mid \lambda \in [0, 1] \right\} \quad \text{with} \quad \Sigma(\lambda) = \frac{\lambda \Sigma_1 + (1 - \lambda)\Sigma_2 + \gamma(\lambda)\Sigma_3}{1 + \gamma(\lambda)} \ , \tag{14}$$

where $\gamma(\lambda) = \frac{4}{5} \exp\left(-(6\lambda - 3)^2\right)$.

For more information on the training setup, we refer the readers to the code.

### A.4.3 MANIFOLDS IN IMAGE LATENT SPACES

**MNIST** We use the standard implementation and data provided by scikit-learn (Pedregosa et al., 2011) with standard train and validation split. A convolutional residual (He et al., 2016) variational autoencoder (Rezende et al., 2014; Kingma & Welling, 2014) architecture with batch norm (Ioffe & Szegedy, 2015) compresses the pixel space into a 20-dimensional latent space. It is trained for approximately 7000 epochs. The resulting VAE gives – to the human eye – perfect reconstructions; one might consider 20 dimensions even too many to describe the space that MNIST digits live in. As a result, it is the Marginal Flow's task to find conditional lower-dimensional manifolds that describe the 20-dimensional latent space well. In our experiments, we fit both a 1- (Figure 10) and a 2-dimensional manifold (Figure 15) with a uniform base distribution $p_{\text{base}} = \mathcal{U}([-1, 1])^d$ with $d = \{1, 2\}$. The label information is one-hot encoded. We train Marginal Flow with $N_c 0256$ for 300 epochs. The neural network $f_\theta(\boldsymbol{z})$ has 3 layers with 256 neurons each.

**JAFFE** We use $64 \times 64$ px crops to the face area. We split the data into 80% training and 20% validation set. The convolutional residual variational autoencoder compresses the images into a 10-dimensional space. After training for about 9000 epochs, there is no visible reconstruction error. The values for happiness, sadness, surprise, anger, disgust and fear are continuous float values and are provided to the Marginal Flow as conditioning parameter $\boldsymbol{c}$. For a neutral facial expression, we set all values to the minimum value found in the dataset (around 1.1). For a medium level of an emotion, we set that value to 3.0 while leaving all other emotions at minimum value. The same goes

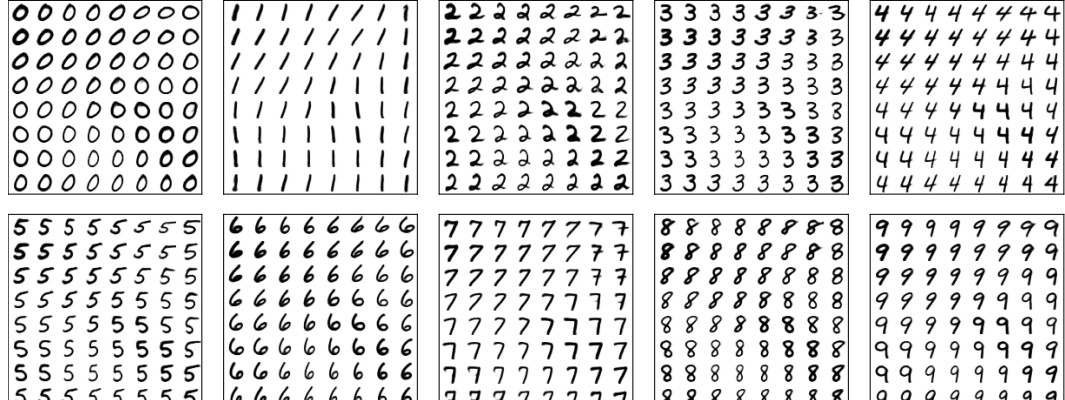

Figure 15: Marginal Flow trained with 2-dim base distribution on 20-dim MNIST latent space. We show the learned 2-dim manifold conditioned on the class label.

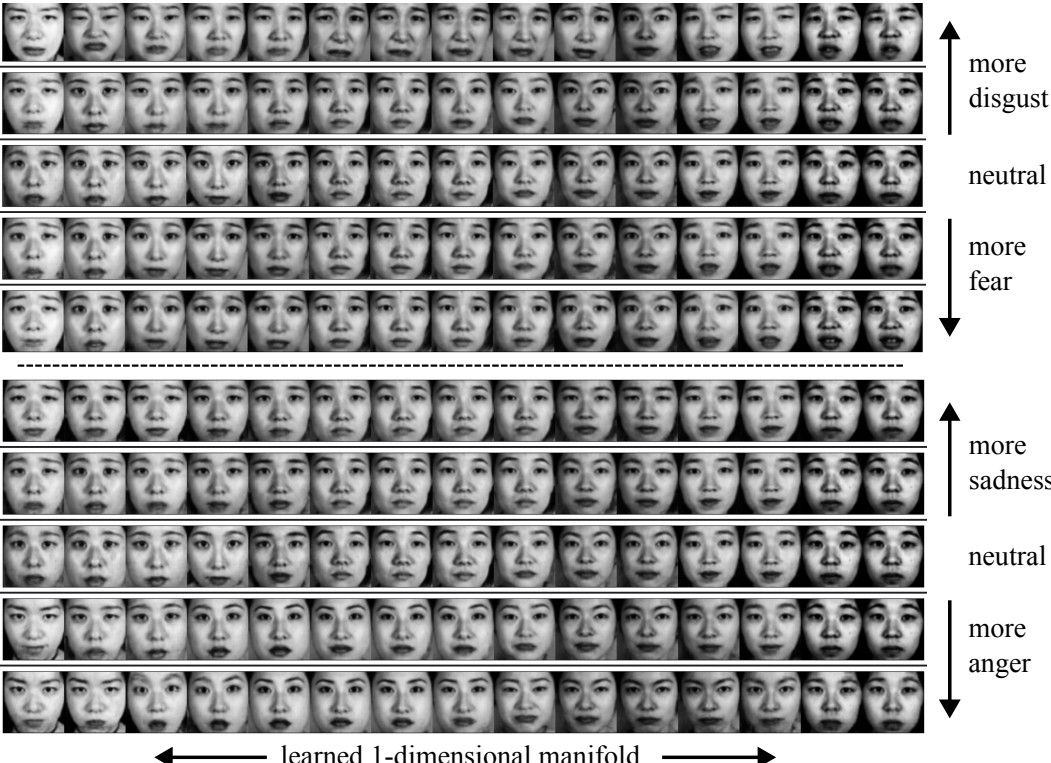

learned 1-dimensional manifold

Figure 16: The JAFFE dataset provides images and labels for the emotions happiness, and surprise (see main text), and further sadness, anger, disgust, and fear. Here, we show results images for generating images with conditioning for the latter four emotions.

for a high level of that emotion with the value being $4.8$, the maximum found in the dataset. We train the Marginal Flow with $N_c = 128$ for 300 epochs. The neural network $f_\theta(z)$ has 3 layers with 256 neurons each.

### A.5 VARIANCE AND BIAS OF KL-ESTIMATOR

We consider the marginal in Eq. 1 with the parametrization proposed for our model in Eq. 2, which we rewrite with an explicit dependency on $z$:

$$q_\theta(\boldsymbol{x}) = \mathbb{E}_{\boldsymbol{z}\sim p_{\text{base}}}\big[q(\boldsymbol{x}\mid f_\theta(\boldsymbol{z}))\big] = \int q(\boldsymbol{x}\mid f_\theta(\boldsymbol{z}))\,p_{\text{base}}(\boldsymbol{z})\,d\boldsymbol{z}, \tag{15}$$

with $\boldsymbol{z}\in\mathbb{R}^m$, $\boldsymbol{x}\in\mathbb{R}^d$ and $m<d$. We assume the data to be generated from a lower-dimensional manifold (of intrinsic dimension $m$). For illustrative purposes, we consider the reverse KL divergence:

$$\mathcal{L}(\theta) = \text{KL}(q_\theta\,\|\,p) = \mathbb{E}_{\boldsymbol{x}\sim q_\theta}\big[\log\mathbb{E}_{\boldsymbol{z}\sim p_{\text{base}}}\big[q(\boldsymbol{x}\mid f_\theta(\boldsymbol{z}))\big] - \log p(\boldsymbol{x})\big]. \tag{16}$$

In practice, we approximate both the expectation over $\boldsymbol{x}$ and the inner expectation defining $q_\theta(\boldsymbol{x})$ via Monte Carlo. Given i.i.d. outer samples $\boldsymbol{x}_1,\ldots,\boldsymbol{x}_N\sim q_\theta$ and i.i.d. inner samples $\boldsymbol{z}_1,\ldots,\boldsymbol{z}_{N_c}\sim p_{\text{base}}$, we form the nested Monte Carlo approximation:

$$\widehat{\mathcal{L}}_{N,N_c}(\theta) := \frac{1}{N}\sum_{j=1}^{N}\log\frac{\widehat{q}_{\theta,N_c}(\boldsymbol{x}_j)}{p(\boldsymbol{x}_j)} \quad\text{with}\quad \widehat{q}_{\theta,N_c}(\boldsymbol{x}) := \frac{1}{N_c}\sum_{i=1}^{N_c}q(\boldsymbol{x}\mid f_\theta(\boldsymbol{z}_i)). \tag{17}$$

**Theorem A.1** (Hong & Juneja (2009) Adapted). *Consider a generic nested expectation $I = \mathbb{E}_{\boldsymbol{x}}\big[h(\mu(\boldsymbol{x}))\big]$ with $\mu(\boldsymbol{x}) = \mathbb{E}_{\boldsymbol{z}}\big[y(\boldsymbol{x},\boldsymbol{z})\mid\boldsymbol{x}\big]$, and the nested Monte Carlo estimator*

$$\widehat{I}_{N,N_c} = \frac{1}{N}\sum_{j=1}^{N}h\big(\widehat{\mu}_{N_c}(\boldsymbol{x}_j)\big), \qquad \widehat{\mu}_{N_c}(\boldsymbol{x}_j) = \frac{1}{N_c}\sum_{i=1}^{N_c}y(\boldsymbol{x}_j,\boldsymbol{z}_i),$$

*where $\boldsymbol{x}_j$ are i.i.d. outer samples and $\boldsymbol{z}_i$ are i.i.d. inner samples. If $h(\cdot)$ is three times differentiable and the inner estimator is sufficiently regular, then:*

$$\text{Bias}(\widehat{I}_{N,N_c}) = \frac{1}{2N_c}\mathbb{E}_{\boldsymbol{x}}\big[h''(\mu(\boldsymbol{x}))\,\text{Var}_{\boldsymbol{z}}(y(\boldsymbol{x},\boldsymbol{z})\mid\boldsymbol{x})\big] + \mathcal{O}\left(\frac{1}{N_c^2}\right), \tag{18}$$

$$\text{Var}(\widehat{I}_{N,N_c}) = \frac{\text{Var}_{\boldsymbol{x}}(h(\mu(\boldsymbol{x})))}{N} + \frac{1}{NN_c}\mathbb{E}_{\boldsymbol{x}}\big[(h'(\mu(\boldsymbol{x})))^2\text{Var}_{\boldsymbol{z}}(y(\boldsymbol{x},\boldsymbol{z})\mid\boldsymbol{x})\big] + \mathcal{O}\left(\frac{1}{NN_c^2}\right). \tag{19}$$

**Lemma A.2.** *The Monte Carlo estimate of the KL divergence $\widehat{\mathcal{L}}_{N,N_c}(\theta)$ is characterized by:*

$$\text{Bias}(\widehat{\mathcal{L}}_{N,N_c}) = -\frac{1}{2N_c}\mathbb{E}_{\boldsymbol{x}\sim q_\theta}\left[\frac{\text{Var}_{\boldsymbol{z}}(q(\boldsymbol{x}|f_\theta(\boldsymbol{z})))}{q_\theta^2(\boldsymbol{x})}\right] + \mathcal{O}\left(\frac{1}{N_c^2}\right), \tag{20}$$

$$\text{Var}(\widehat{\mathcal{L}}_{N,N_c}) = \frac{V_{data}}{N} + \frac{1}{NN_c}\mathbb{E}_{\boldsymbol{x}\sim q_\theta}\left[\frac{\text{Var}_{\boldsymbol{z}}(q(\boldsymbol{x}|f_\theta(\boldsymbol{z})))}{q_\theta^2(\boldsymbol{x})}\right] + \mathcal{O}\left(\frac{1}{NN_c^2}\right), \tag{21}$$

*where $V_{data} = \text{Var}_{\boldsymbol{x}\sim q_\theta}(\log q_\theta(\boldsymbol{x}) - \log p(\boldsymbol{x}))$.*

*Proof.* We apply Theorem A.1 with $h(t) = \log(t)$. The derivatives are $h'(t) = 1/t$ and $h''(t) = -1/t^2$. Crucially, we observe that $(h'(t))^2 = 1/t^2$ and $|h''(t)| = 1/t^2$. Therefore, the coefficient for the bias term and the coefficient for the secondary variance term are identical in magnitude: $\frac{1}{\mu(\boldsymbol{x})^2}\text{Var}(y|\boldsymbol{x})$. Substituting $\mu(\boldsymbol{x}) = q_\theta(\boldsymbol{x})$ yields the result. $\square$

### A.6 SCALING WITH DIMENSIONALITY

We investigate the behavior of the bias and variance of the nested Monte Carlo estimator $\widehat{\mathcal{L}}_{N,N_c}$ as a function of the ambient data dimension $d$ and the intrinsic manifold dimension $m$, where $m\ll d$.

**Theorem A.3** (Dimensionality Dependence). *Under Assumptions 1, 2, and 3, the bias and the variance inflation due to the nested estimator depend exclusively on the intrinsic dimension $m$ and are independent of the ambient dimension $d$. While the intrinsic data variance $V_{data}$ scales linearly with $d$, the Monte Carlo penalty term does not. Specifically, assuming for tractability that the latent posterior is centered at the prior mean ($z^* = 0$), we have:*

$$\text{Bias}(\widehat{\mathcal{L}}_{N,N_c}) = -\frac{\gamma}{2N_c} + \mathcal{O}\left(\frac{1}{N_c^2}\right), \tag{22}$$

$$\text{Var}(\widehat{\mathcal{L}}_{N,N_c}) = \frac{V_{data}}{N} + \frac{\gamma}{NN_c} + \mathcal{O}\left(\frac{1}{NN_c^2}\right), \tag{23}$$

*where $\gamma = \left(\frac{(1+\sigma^{-2})^2}{1+2\sigma^{-2}}\right)^{m/2} - 1$ is a constant depending only on $m$ and $\sigma$.*

*Proof.* We analyze the Squared Coefficient of Variation, defined as $\text{CV}^2 = \frac{\mathbb{E}_z[L(z)^2]}{(\mathbb{E}_z[L(z)])^2} - 1$, where $L(z) = q(x \mid f_\theta(z))$ is the conditional likelihood. Using Assumptions 2 and 3, and the property $J^\top J = I_m$, the likelihood factorizes into a component orthogonal to the manifold and a component parallel to it:

$$L(z) = \underbrace{\frac{e^{-\|x_\perp\|^2/2\sigma^2}}{(2\pi\sigma^2)^{d/2}}}_{C_{\text{orth}}(x_\perp, d)} \cdot \underbrace{e^{-\|z-z^*\|^2/2\sigma^2}}_{K_{\text{par}}(z,z^*)}. \tag{24}$$

The term $C_{\text{orth}}$ encapsulates all dependencies on the ambient dimension $d$ (via the normalization constant) and the reconstruction error $\|x_\perp\|$. Since $C_{\text{orth}}$ is constant with respect to $z$, it cancels out in the ratio of expectations:

$$\frac{\mathbb{E}_z[L(z)^2]}{(\mathbb{E}_z[L(z)])^2} = \frac{C_{\text{orth}}^2 \mathbb{E}_z[K_{\text{par}}(z,z^*)^2]}{C_{\text{orth}}^2(\mathbb{E}_z[K_{\text{par}}(z,z^*)])^2} = \frac{\mathbb{E}_z[K_{\text{par}}(z,z^*)^2]}{(\mathbb{E}_z[K_{\text{par}}(z,z^*)])^2}. \tag{25}$$

This ratio now depends solely on $m$-dimensional quantities. For the explicit calculation of $\gamma$, we consider the case where the optimal code lies at the prior mean, $z^* = 0$. We compute the expectations against the prior $p(z) = \mathcal{N}(0, I_m)$:

$$\mathbb{E}_z[K_{\text{par}}] \propto \int e^{-\frac{1}{2}\|z\|^2} e^{-\frac{1}{2\sigma^2}\|z\|^2} dz = \int e^{-\frac{1}{2}(1+\sigma^{-2})\|z\|^2} dz \propto (1+\sigma^{-2})^{-m/2}, \tag{26}$$

$$\mathbb{E}_z[K_{\text{par}}^2] \propto \int e^{-\frac{1}{2}\|z\|^2}\left(e^{-\frac{1}{2\sigma^2}\|z\|^2}\right)^2 dz = \int e^{-\frac{1}{2}(1+2\sigma^{-2})\|z\|^2} dz \propto (1+2\sigma^{-2})^{-m/2}. \tag{27}$$

Substituting the normalization constants $(1+\sigma^{-2})^{m/2}$ and $(1+2\sigma^{-2})^{m/2}$ appropriately into the ratio yields $\gamma = \frac{(1+\sigma^{-2})^m}{(1+2\sigma^{-2})^{m/2}} - 1$. *Remark on $z^* \neq 0$:* If the code is not centered ($z^* \neq 0$), the convolution of the Gaussian prior and likelihood induces a shift term depending on $\|z^*\|$. However, this term remains strictly a function of the intrinsic geometry ($z \in \mathbb{R}^m$) and does not re-introduce any dependence on the ambient dimension $d$. Thus, the independence from $d$ holds generally. $\square$

### A.7 EMPIRICAL VALIDATION

We perform a simulation to verify the results for a target distribution on a manifold. For this, the true analytical reverse KL is required. As such, we use a simple target multivariate distribution. Consider two independent random variables:

$$X = X_1 + X_2, \qquad X_1 \sim \mathcal{N}(0, \sigma_1^2 I_d), \qquad X_2 = \mu + Az, \qquad z \sim \mathcal{N}(0, \sigma_2^2 I_m), \tag{28}$$

While $X_1$ is a simple isotropic Gaussian, $X_2$ is a linear mapping of another Gaussian with dimension $m < d$, using $A \in \mathbb{R}^{d \times m}$ and the mean $\mu \in \mathbb{R}^d$. With these two independent variables, we can conclude

$$\mathbb{E}[X] = \mu, \quad \text{Var}[X] = \sigma_1^2 I_d + \sigma_2^2 AA^T. \tag{29}$$

Alternatively, this means $X \sim \mathcal{N}(\mu, C)$ where $C = \sigma_1^2 I_d + \sigma_2^2 AA^T$. As such, we can express the distribution of $X$ directly as a multivariate normal or an integral over a subspace described by $\mu$ and $A$:

$$\mathcal{N}(\mu, C) = \int \mathcal{N}(x; \mu + Az, \sigma_1^2 I_d)\mathcal{N}(z; 0, \sigma_2^2 I_r)dz = \int q(x \mid w)q(w)dw. \tag{30}$$

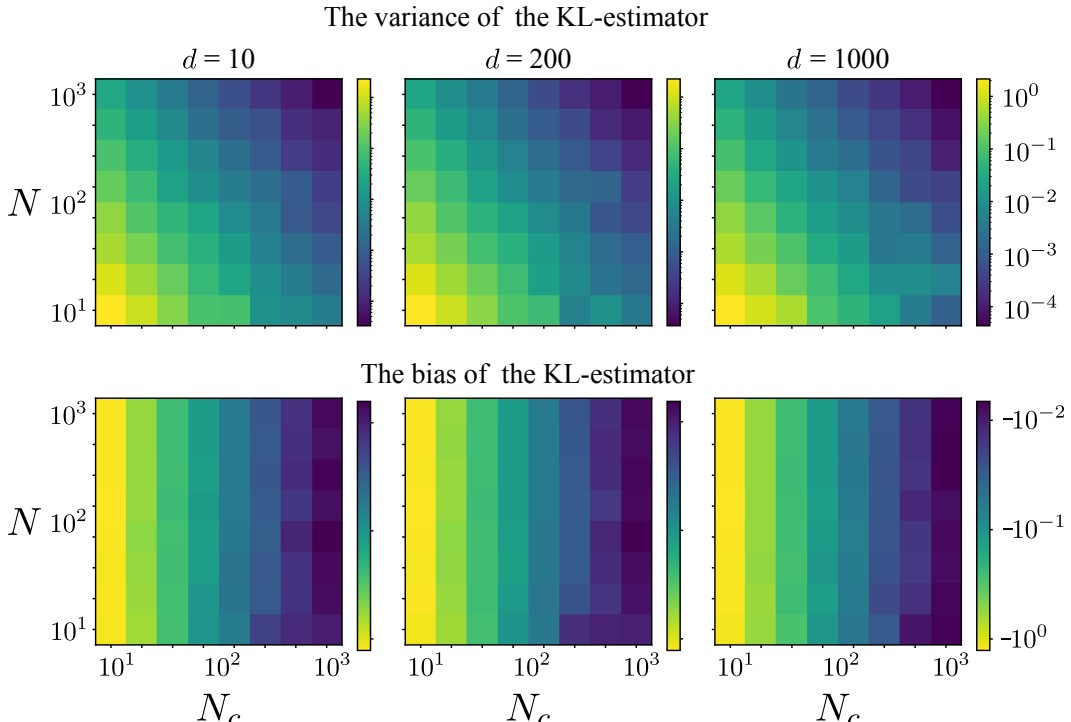

Figure 17: The variance and bias of the Monte Carlo estimator $\widehat{\mathcal{L}}_{N,N_c}$ of the reverse KL divergence for a target distribution with a manifold structure. For each value, 500 simulations are performed.

Intuitively, this formulation allows us to circumvent the training procedure by knowing the true manifold $q(w)$. Avoiding any training error results in $\mathcal{L} = 0$. With this, we can verify our theoretical results for the bias and variance of $\widehat{\mathcal{L}}$. The integral over the subspace is used as $q(w)$, and the multivariate normal corresponds to the target $p$. To showcase the manifold hypothesis, we use the extreme case $r = 1$, leading to a rank-1 update to the covariance of the isotropic $X_1$. Figure 17 shows the variance and bias for various dimensions $d$. The variance and bias behave as predicted in Equation 23 and even show no dependence on $d$.

