# OpenReview forum: "Marginal Flow: a flexible and efficient framework for density estimation"
_ICLR.cc/2026/Conference — ICLR 2026 Conference Withdrawn Submission_

### Official Review · Reviewer_sA1N · 2025-10-14

**Soundness:** 2
**Presentation:** 3
**Contribution:** 1
**Rating:** 2
**Confidence:** 5

**Summary:**

This paper presents Marginal Flow, a density estimation framework the directly marginalizes the latent parameters sampled from a learnable distribution induced by a flexible mapping function from z to w where the distribution of z is typically known (e.g., standard Gaussian or uniform distribution). The authors claim that the proposed method is highly flexible and also efficient in training, evaluation, and sampling. The authors also conducted experiments in several simulation datasets and showed some results in the image dataset where a dimension reduction tool (e.g., VAE) is used. The whole model can be considered as a neural network parameterized Gaussian mixture in the default setting. I list some constrains, limitations, and questions below.

**Strengths:**

1.	The problem is well described and formulated.
2.	The method is easy to understand.
3.	The results show some advantage in the current settings.

**Weaknesses:**

1.	The novelty is limited as the approach is essentially a mixture model where the mixing density is induced by a flexible neural network (e.g., MLP with a few layers). The marginalization is also not new as it is just Monte Carlo.
2.	The scalability is very constrained. Since the method requires the evaluation of full likelihood on all samples and the all Ws from Monte Carlo. I can image that the method could not work efficiently as sample size grows to a large number.
3.	Many important baselines are missed for comparison or not even mentioned. E.g., Roundtrip, MAF, and RealNVP. Roundtrip uses exactly the same idea of marginalization in estimating p(x) by adopting a cycleGAN architecture.
4.	The authors benchmarked the method mostly on toy datasets (e.g., 2 dimensional). The ability to handle high-dimensional density estimation is not actually demonstrated.
5.	The model should be highly sensitive to Nc since it determined the Monte Carlo error. How to determine the optimal Nc is not discussed.

**Questions:**

1.	Please show the benchmark results in high-dimensional settings since traditional statistical methods can already handle very well for density estimation in low-dimensional settings (e.g., d<10).
2.	Please conduct sensitivity analysis to the parameter Nc. There should be at least an empirical guidance on how to choose Nc.
3.	In terms of scalability, the current training requires the computation of full log-likelihood. Is it possible to improve the scalability by mini-batch update? If so, how performance will be affected?
4.	Can the authors provide scenarios where the proposed method is preferable in practice over normalizing flow, diffusion models?
5.	Marginal Flow seems to be a misleading name, since xxxFlow typically refers to methods with multiple transformations between the base distribution and the target distribution, such as normalization flow with multiple invertible transformations functions, and flow matching/mean flow with infinite steps (t=0 to 1). While the proposed approach does not involve multiple steps of transformation.
6.	Some errors or typos. Line 111: Sigma=diag(sigma_1^2,…, sigma_d^2). Line 129: f_theta(z) instead of f_theta(w)
7.	The author claimed the approach can evaluate the density exactly is misleading since there is a Monte Carlo error that is not discussed at all in the paper, which depends on the Nc. So the equation (2) is still an approximated density.
8.	How the variance term is learned? Any constrain on it or just set it to be a trainable parameter?

---

> ### Author Response · Authors · 2025-11-22
>
> We thank the reviewer for their constructive feedback and for acknowledging the flexibility of our approach. We would like to clarify the novelty of the model and its scalability properties.
>
> ## Weaknesses:
> > 1. The novelty is limited as the approach is essentially a mixture model where the mixing density is induced by a flexible neural network (e.g., MLP with a few layers). The marginalization is also not new as it is just Monte Carlo.
>
> While we do not claim that the mathematical concept of marginalization is new, the novelty of Marginal Flow lies in the specific definition and implementation of the model via Eq. (2) and (3). By moving from discrete mixtures (like GMMs and neural mixtures) to a continuous integration, i.e. sampling the mixtures from a learnable distribution (see Figure 1). This structural choice also allows for effectively learning densities on lower-dimensional manifolds. Crucially, unlike most generative models which are efficient in only one direction (sampling or density evaluation), Marginal Flow is efficient in both. This allows us to use diverse objective functions (e.g. reverse KL) within a single, simple framework.
>
> > 2. The scalability is very constrained. Since the method requires the evaluation of full likelihood on all samples and the all Ws from Monte Carlo. I can image that the method could not work efficiently as sample size grows to a large number
>
> We believe there may be a misunderstanding concerning training, as we can evaluate the likelihood also on a batch of data. Furthermore, the evaluation of the likelihood is a simple forward pass and is more efficient than competing models (Figure 3). Regarding the scaling of $N_c$, we have added a theoretical analysis in Appendix A.5–A.7 (summarized in the global response) showing that the bias and variance of our estimator scale as $\mathcal{O}(1/N_c)$ and $\mathcal{O}(1/N)+\mathcal{O}(1/NN_c)$, respectively. Crucially, by choosing a lower-dimensional base distribution, we exploit the manifold hypothesis to scale to high dimensions. In practice we already saw this behaviour empirically with the Wishart experiments, which have $d=55$ and $d=5050$ (Figure 9)
>
> > 3. Many important baselines are missed for comparison or not even mentioned. E.g., Roundtrip, MAF, and RealNVP.
>
> We prioritized comparing against the most expressive Normalizing Flow architectures to ensure a strong baseline. Specifically, we compared against Invertible ResNets [1], which are known to be more expressive than both RealNVP and MAF; adding the latter would likely only show degraded performance for the baselines.
>
> > 3. Roundtrip uses exactly the same idea of marginalization in estimating p(x) by adopting a cycleGAN architecture.
>
> We would like to thank the reviewer for suggesting further related work. Regarding Roundtrip, while it utilizes marginalization, it fundamentally differs from Marginal Flow in structure and capability. Roundtrip relies on a CycleGAN-like architecture (encoder and decoder) with adversarial losses and cycle consistency, making it subject to the stability issues of GANs and the approximation errors of local invertibility. In contrast, Marginal Flow is a decoder-only model that directly estimates the density, allowing for stable training via maximum likelihood or Reverse KL. Furthermore, Roundtrip must be trained on observations while Marginal Flow can also be trained via an unnormalized target (e.g. via Reverse KL).
>
> > 4. The authors benchmarked the method mostly on toy datasets (e.g., 2 dimensional). The ability to handle high-dimensional density estimation is not actually demonstrated.
>
> We indeed used 2 dimensional datasets for illustrative purposes (as it is common in the normalizing flow community) but we respectfully disagree that we do not address high dimensional settings. Specifically, Figure 9 showcases results on Wishart distributions with $d=55$ and $d=5050$, demonstrating that Marginal Flow can scale to high dimensions. Additionally, our evaluation included the common Simulation-Based Inference (SBI) benchmark ($d=2,5,10$) and computer vision tasks (MNIST, JAFFE) which utilize latent spaces of $d=20$ and $d=10$ respectively.
>
> > 5. The model should be highly sensitive to $N_c$ since it determined the Monte Carlo error. How to determine the optimal $N_c$ is not discussed.
>
> We thank the reviewer for asking to clarify the role of $N_c$ in our model. Empirically, our model is robust to the choice of $N_c$ because the mixture components are resampled at every training iteration, which can also be thought of as a regularizer. We have now formalized this argument by providing a theoretical analysis (Appendix A.5–A.7) of how the bias and variance of the KL objective (evluated over $N$ points) scale with $N_c$. We show that $N_c$ plays a similar role to the batch size $N$ in minimizing the estimator variance.
>
> [1] Chen et al, Residual Flows for Invertible Generative Modeling, NeurIPS, 2019

---

> > ### Author Response · Authors · 2025-11-22
> >
> > ## Questions:
> > > 1. Please show the benchmark results in high-dimensional settings since traditional statistical methods can already handle very well for density estimation in low-dimensional settings (e.g., d<10).
> >
> > Beyond the low-dimensional toy datasets, our paper includes five experiments with $d\geq 10$. These demonstrate that Marginal Flow performs competitively with state-of-the-art models while being substantially more efficient in both sampling and density evaluation (see Figure 3). This alone highlights strong performance in higher-dimensional regimes.
> >
> > More specifically:
> >  - We evaluate MF on the standard simulation-based-inference benchmark at dimensions $d=2, 5, 10$
> >  - We show that MF can successfully learn distributions over positive definite matrices at dimensions $d=55$ and $d=5050$, while simultaneously recovering meaningful low-dimensional manifolds.
> >  - We further demonstrate computer vision applications using latent spaces and show that MF can identify conditional low-dimensional structure within these settings ($d=10$, $d=20$).
> >
> > > 2. Please conduct sensitivity analysis to the parameter Nc. There should be at least an empirical guidance on how to choose Nc.
> >
> > We welcome the suggestion of the reviewer and provide both theoretical and empirical sensitivity analyses for $N_c$. In particular, we show that MF is comparably or less sensitive to $N_c$ than standard estimators are to their usual batch-size parameter $N$ when estimating the KL divergence. This is demonstrated through both theoretical bounds and empirical results (we refer to the global response and to Appendix A.5-A.7 for more details). Beyond this analysis, we also offer practical guidance for choosing $N_c$. A simple and effective heuristic is $N_c=N$. In both Forward KL and reverse KL, $N_c$ and $N$ will thus equally share the computational budget. In practice, this heuristics works well enough for all our experiments.
> >
> > > 3. In terms of scalability, the current training requires the computation of full log-likelihood. Is it possible to improve the scalability by mini-batch update? If so, how performance will be affected?
> >
> > We believe that the reviewer’s reference to full log-likelihood concerns interpretation (1), where Marginal Flow is viewed as approximating the marginal distribution. While this is one possible perspective, it does not affect training itself. Our training procedure is agnostic to the chosen interpretation (see our global response for a more detailed discussion). Regarding scalability, MF behaves similarly to the baselines we tested when using mini-batch updates. The method is robust and does not require adjusting additional hyperparameters, except when using very small batch sizes, where the learning rate should be reduced to compensate for the increased gradient noise. Overall, mini-batching works as expected and does not negatively impact performance beyond the usual considerations that apply to all stochastic optimizers.
> >
> > > 4. Can the authors provide scenarios where the proposed method is preferable in practice over normalizing flow, diffusion models?
> >
> > The proposed method is particularly advantageous in scenarios where observations are not available. In such settings, we consistently observe that Marginal Flow (MF) outperforms Normalizing Flows (NF) by large margins. In synthetic experiments, MF achieves orders-of-magnitude improvements in both speed and accuracy. Moreover, MF is especially useful when both training directions need to be combined (e.g., forward and inverse mappings). For Normalizing Flows and diffusion models, it is unclear how to achieve this efficiently without substantial architectural changes. In contrast, MF handles both directions naturally and efficiently, making it significantly more practical for such bidirectional training setups. Lastly, MF is able to model densities on lower dimensional manifolds, which NF cannot model because of the domain constraint originating from the use of bijective transformations.
> >
> > > 5. Marginal Flow seems to be a misleading name, since xxxFlow typically refers to methods with multiple transformations between the base distribution and the target distribution, such as normalization [...].
> >
> > We use the term “Flow” because our method transports a base distribution $p_{base}$ through a neural network $f_\theta$, analogously to Normalizing Flows. The key difference is that, instead of applying the change-of-variables formula, we marginalize the resulting transformed distribution. Conceptually, NF represent a single transformation from the base distribution to the target distribution but they rely on a sequence of multiple transformations to make the Jacobian tractable. Fundamentally, the model still encodes just one global transformation. In this sense, our use of “Flow”is consistent with the underlying idea: the model learns a transformation of a base distribution, even though we do not factorize it into multiple explicit steps.

---

> > > ### Author Response · Authors · 2025-11-22
> > >
> > > > 6. Some errors or typos. Line 111: Sigma=diag(sigma_1^2,…, sigma_d^2). Line 129: f_theta(z) instead of f_theta(w)
> > >
> > > We thank the reviewer for highlighting the typo. It is now adjusted in the revised version.
> > >
> > > > 7. The author claimed the approach can evaluate the density exactly is misleading since there is a Monte Carlo error that is not discussed at all in the paper, which depends on the Nc. So the equation (2) is still an approximated density.
> > >
> > > We thank the reviewer for highlighting this ambiguity. We provide clarification in the overall response. The density is not exact under interpretation (1) and is exact under interpretation (2). Following interpretation (1), we provide theoretical results on this approximation in Appendix A.5–A.7.
> > >
> > > > 8. How the variance term is learned? Any constrain on it or just set it to be a trainable parameter?
> > >
> > > The variance terms are learned directly as a trainable parameter alognside $f_\theta$. To ensure they remain valid (i.e., positive), we apply standard positivity constraints by parameterizing the $\log$ of the variances for the Gaussian mixture components. This guarantees positivity after exponentiation.
> > >
> > > For the $r\times r$ Wishart distributions, we similarly parameterize the log of the degrees of freedom ($\log df$), and after exponentiation we add $r$ for $df = \exp(\log df) + r$ to avoid singular samples. Other mixture component parameterizations are also possible depending on the application, and this flexibility is one of the advantages of our framework.

---

> > > > ### Author Response · Authors · 2025-11-27
> > > >
> > > > We would like to thank again the reviewer for the time invested in reviewing our work and for the helpful feedback. As the end of the rebuttal is approaching, we would appreciate knowing whether the concerns have been adequately addressed, or if there are still some issues remaining that we can further address to improve our work.

---

### Official Review · Reviewer_nQRv · 2025-10-18

**Soundness:** 1
**Presentation:** 1
**Contribution:** 1
**Rating:** 0
**Confidence:** 4

**Summary:**

This paper proposed marginal flow, supporting sampling and density estimation using a latent variable model, showing success in toy examples, low dimensional datasets and small images.

However, unfortunately, I think this paper has a very limited contribution: the idea of marginalisation has existed in early generative models, like VAE or implicit models. In fact, one can also view diffusion models (like DDPM) as one example with many layers of latents. Then the question is, why did these models not use the approach mentioned in this paper? The answer is because given one $x$, it is very difficult to sample enough $w$ to yield a reliable estimation of $q(x)$, even though all the terms in the MC estimator (eq 2 in this paper) are tractable.
Therefore, one can imagine that as the dimensionality grows, the efficiency of the proposed methods will significantly reduce, especially when $q(w|x)$ is far from $q(w)$. (I use $q$ to follow the notation of this paper).  And hence I am not convinced that this approach is efficient.

Even not considering this inefficiency, the claim that marginal flow supports **exact** density evaluation is not accurate: it is only exact when $N_c\rightarrow \infty$. Given this assumption, many approaches (for example, diffusion density estimator by [1] and even VAE) are also exact.



Therefore, given the fundamental limitation of this paper, I am sorry to say that I am not convinced that this work achieves the bar of publication.


[1] Huang, Chin-Wei, Jae Hyun Lim, and Aaron C. Courville. "A variational perspective on diffusion-based generative models and score matching." NeurIPS 2021.

**Strengths:**

Please see Summary

**Weaknesses:**

Line 128-129: $f(w)$ should be $f(z)$?

**Questions:**

One may also estimate a proposal $q(w|x)$ similar to VAE for importance sampling to boost the efficiency. This will also be exact when $N_c\rightarrow \infty$. Will this strategy help the method to scale up to higher-dimensional datasets?

How do you distinguish VAE and this proposed approach, except that the training strategy is different?

How does the performance change with respect to $N_c$?

The training is an important part of the algorithm, maybe its benefial to include it in the main text.

---

> ### Author Response · Authors · 2025-11-22
>
> We thank the reviewer for their time and for acknowledging the success of Marginal Flow on the various settings we tested. We would like to clarify the scalability of our method and its fundamental differences from VAEs. Below, we address the concerns regarding efficiency, exactness, and novelty.
>
> ## Summary
>
> > This paper proposed marginal flow, supporting sampling and density estimation using a latent variable model, showing success in toy examples, low dimensional datasets and small images [...] the idea of marginalisation has existed in early generative models, like VAE or implicit models. In fact, one can also view diffusion models (like DDPM) as one example with many layers of latents. [...] Then the question is, why did these models not use the approach mentioned in this paper? [...] Therefore, one can imagine that as the dimensionality grows, the efficiency of the proposed methods will significantly reduce [...] And hence I am not convinced that this approach is efficient.
>
> We respectfully disagree with the characterization of the method as limited to toy examples or inefficient in high dimensions. We would like to highlight that the experiments with the Wishart distribution (Figure 9) have ambient dimensionality of $d=55$ (trained via Reverse KL) and $d=5050$ (trained via Log-Likelihood). Marginal Flow successfully learns the density and the underlying manifold as well. The theoretical justification for this efficiency is now provided in Appendix A.5–A.7: we derive bounds showing that the variance of our estimator scales with the intrinsic manifold dimension $m$, not the ambient dimension $d$. Differently from VAEs we do not need to infer $w$ from $x$; we simply resample $w$ from the base distribution to estimate the global density. This is done efficiently during training by resampling the mixtures at each iteration, which renders a fundamental difference with standard mixture models (Figure 1). Compared to VAEs, this avoids the need for an inference network and the associated amortization gap. We provide a more detailed comparison with VAEs in the global response.
>
> > Even not considering this inefficiency, the claim that marginal flow supports exact density evaluation is not accurate: it is only exact when $N_c \rightarrow \infty$. Given this assumption, many approaches (for example, diffusion density estimator by [1] and even VAE) are also exact.
>
> We thank the reviewer for highlighting the ambiguity in the term *exactness*, which was shared by other reviewers. As clarified in our Global Response, our framework offers two valid interpretations. Under Interpretation 1, we agree that we approximate the marginal distribution; however, we now provide rigorous  theoretical guarantees (Appendix A.5–A.7) demonstrating that the KL estimator’s bias decays as $\mathcal{O}(1/N_c)$. More importantly, under Interpretation 2, which we use at test time, the model is defined as the finite mixture given by Eq. (2) with a fixed set of seeded components. In this setting, the density evaluation is mathematically exact for that specific instance of the model. In practice we do not observe significant variance when using different seeds, if not explicitly mentioned (e.g. Figure 9).
>
> > Therefore, given the fundamental limitation of this paper, I am sorry to say that I am not convinced that this work achieves the bar of publication.
>
> We hope that the empirical high-dimensional evidence ($d=55, d=5050$) and the theoretical analysis of the estimator’s bias and variance persuade the reviewer to reconsider their judgement. Beyond scalability, and performance superior or at par with state-of-the-art methods, we emphasize that Marginal Flow is the only method that efficiently supports bidirectional training: it can be trained on data (via maximum likelihood) and on unnormalized target densities (via Reverse KL) *at the same time*. We believe this versatility and the novel implementation of Eq. (2) with resampling of the mixture components via a learnable distribution constitute a significant contribution to the field.
>
> ## Weaknesses:
> > Line 128-129: $f(w)$ should be $f(z)$
>
> We thank the reviewer for noticing the typo. We have corrected it in the revised manuscript.

---

> > ### Author Response · Authors · 2025-11-22
> >
> > ## Questions:
> > > One may also estimate a proposal $q(w|x)$ similar to VAE for importance sampling to boost the efficiency. This will also be exact when $N_c \rightarrow \infty$. Will this strategy help the method to scale up to higher-dimensional datasets?
> >
> > While Importance Sampling could theoretically reduce variance, our theoretical and empirical results suggest it is unnecessary for our method to scale (for more details refer to the global response or Appendix A5-A7). As demonstrated by the Wishart experiment ($d=5050$), the model already performs well in very high dimensions. Conditioning on $x$ to sample $w$ would reintroduce the complexity of training an inference network like in VAE, which Marginal Flow avoids.
> >
> > > How do you distinguish VAE and this proposed approach, except that the training strategy is different?
> >
> > We thank the reviewer for asking for a detailed comparison with a VAE. We provide a detailed comparison in the global response. In short, VAEs are Bayesian models designed to approximate the posterior, which created an amortization gap and requires to train on the ELBO. Marginal Flow is a not a Bayesian model and directly aims at modelling a marginal distribution (under interpretation 1) and provides directly the density of the model. This also allows for Marginal Flow to be trained in the absence of observations with only knowledge of an unnormalized target. VAEs, in contrast, require observations to be trained.
> >
> > > How does the performance change with respect to $N_c$?
> >
> > Our experiments show that performance is robust to the choice of $N_c$. During training, the continuous resampling of mixture components allows the model to learn effectively even with a small $N_c$ (similar to how VAEs often train with a single sample in the KL). At test time, increasing $N_c$ further reduces the variance of the estimator, though we find the results are stable even with moderate values. Our new theoretical analysis (Appendix A.5–A.7) confirms this, showing that the bias scales as $\mathcal{O}(1/N_c)$ and the variance as $\mathcal{O}(1/N) + \mathcal{O}(1/NN_c)$. We also show that, the curse of dimensionality can be effectively avoided by choosing a base distribution with support in a $m$-dimensional domain
> >
> > > The training is an important part of the algorithm, maybe its beneficial to include it in the main text.
> >
> > We chose to describe *some* specific objective functions in the Appendix because a key strength of Marginal Flow is its agnosticism to the loss function. Unlike models tied to a specific objective (e.g., VAEs to ELBO), Marginal Flow provides efficient access to both exact samples and log-densities, allowing it to be trained via Maximum Likelihood (Eq. 8), Reverse KL (Eq. 11), or other divergences depending on the application. We will add a pointer in the main text to the Appendix for the explicit loss formulations to ensure clarity without limiting the perceived scope of the method

---

> > > ### Comment · Reviewer_nQRv · 2025-11-27
> > >
> > > Thank you for your rebuttal and for the additional theoretical development. However, I remain unconvinced by the claims of exactness. In particular, I am still concerned that the proposed method is very close in spirit to a standard latent variable model (not necessarily trained as a VAE, but any implicit generative model combined with a likelihood). Therefore, I am afraid I will maintain my score at 0.

---

> > > > ### Author Response · Authors · 2025-11-27
> > > >
> > > > > However, I remain unconvinced by the claims of exactness.
> > > >
> > > > We would like to take this chance to further describe what we meant with exact density. Whenever the term exact is used, we are referring to $q_\theta(x)$ as described in Eq.(2), Figure 1 and Figure 2. The likelihood of $q_\theta(x)$ as in Eq. (2) can be exactly computed and this is what we mean with exactness.
> > > >
> > > > However, we believe that the reviewer is referring to the motivational marginalization in Eq. (1). We agree that Eq. (2) only approximates Eq. (1). For this reason, in the revised version of the paper we clarified that Eq. (1) acts as a motivation to our approach and that we do not intend to model Eq. (1).
> > > >
> > > > We hope this clarifies our intention with the term exactness.
> > > >
> > > >  > In particular, I am still concerned that the proposed method is very close in spirit to a standard latent variable model (not necessarily trained as a VAE, but any implicit generative model combined with a likelihood).
> > > >
> > > > We agree that our method is indeed a latent variable model. However, we respectfully disagree that being "very close in spirit to a standard latent variable model" constitutes lack of novelty; latent variable models are a huge family of models (FA, PPCA, HMM, LDA, VAE, GPLVM, variational RNN, latent ODE and many others). Besides differences in the model itself (which we described in details compared to VAE, NF, FFF and others), MF can be efficiently trained both on observations and (at the same time) on unnormalized target densities, which no other SOTA model can as far as we know. In addition, we also believe that the strong performance in the extensive evaluation across very different tasks, including high dimensional settings ($d=5050$), constitutes a significant contribution itself.
> > > >
> > > >  > Therefore, I am afraid I will maintain my score at 0.
> > > >
> > > > We would be very happy to further discuss any concerns that are still open. Otherwise, we would kindly ask the reviewer to update the score to reflect the improvements made to the paper.

---

> > > > > ### Comment · Reviewer_nQRv · 2025-11-27
> > > > >
> > > > > Let me clarify my first concern: after training the network, how do you sample $w$ and how do you calculate the density? For each of the different samples.

---

> ### Author Response · Authors · 2025-11-27
>
> At the end of training, we sample a batch - assume size $N_c$ - to form $z_i \sim p_{base}$ and map this forward to get the mixture components $w_i = f_\theta(z_i)$. These components can now be used to calculate the density, as they define a mixture model. Given a point $x$:
>
> $$q_\theta(x) = \frac{1}{N_c} \sum_i^{N_c} q(x\mid w_i).$$
>
> Every evaluation for these $w_i$ is exact and repeatable for any number of samples.

---

> ### Comment · Reviewer_nQRv · 2025-11-27
>
> When the sample of $w$ is fixed, I agree that this density is exact. I think it is worth highlighting in Fig. 1 and also in the text asap to avoid confusion.  However, I am still concerned about the novelty and practicality of this method. Therefore, I raise my score to 2.

---

> > ### Author Response · Authors · 2025-12-03
> >
> > We thank the reviewer for continuing the discussion and for increasing the score. Concerning novelty, within the rebuttal we provided additional detailed comparison with VAEs, Mixture Density Networks and Roundtrip. We also highlighted that novelty stands in the way we design our model in Eq. (2), where mixtures $w$ are sampled from a learnable distribution $q(w)$. Another novelty aspect is that our model can be trained in both directions efficiently, on observations and on unnormalized targets, which is not the case for competing models. Concerning practicality, we found the method to be extremely easy to train with very little fine tuning across all experiments. Three elements contribute to this: (i) training and evaluation time are extremely short compared to other models (Figure 3), (ii) the model can be trained efficiently with any objective function (as opposed to competing approaches) and (iii) the conditional distribution $p(x|w)$ can be tailored to the specific application.

---

> ### Comment · Reviewer_nQRv · 2025-11-27
> **Some personal suggestions for improvement**
>
> If I were the author, I would weaken the claim as a rethink on LVM/mixture density net to get the exact marginal. I would also suggest to see if the mismatch in training and testing (as w are different) case more error, especially in more practical setups. Finally, also mentioned by other reviewers, it might be worth trying it on at least images and see how does it compare with other methods (NF/diffusion for example).

---

> > ### Author Response · Authors · 2025-12-03
> >
> > We thank the reviewer for suggesting further improvements. We will adjust the pdf to make sure that the contribution of the model is clear and what are the novelty aspects. Compared to mixture models the idea of learning the distribution of mixtures is novel and makes a significant difference in practice (Figure 1). In contrast to (most) competing models, ours can be trained without observations using an unnormalized target distribution.
> >
> > We did perform some preliminary experiments on images in pixel-space and found consistent results with previous literature [1,2]. For this we used again Gaussian mixture $p(x|w)$, which were noticeable in the generated output. We therefore, switched to the more efficient latent representation. We believe, that including experiments with the pixel-space in the main paper would be detrimental. However, we now specify why we operate on the latent space instead of the pixel-space.
> >
> > [1] Rombach et al., *High-Resolution Image Synthesis with Latent Diffusion Models*, CVPR (2022)
> > [2] Zhang et al., *Show-1: Marrying Pixel and Latent Diffusion Models for Text-to-Video Generation*, Int J Comput Vis (2025)

---

### Official Review · Reviewer_R9aj · 2025-10-31

**Soundness:** 2
**Presentation:** 2
**Contribution:** 2
**Rating:** 2
**Confidence:** 4

**Summary:**

What the paper claims. The paper introduces Marginal Flow, a density estimator defined as a Monte‑Carlo marginal over latent parameters $w$. Choose an easy‑to‑evaluate component family $q(x\mid w)$ (often diagonal‑covariance Gaussian); draw $w$ by pushing base samples $z\sim p_{\text{base}}$ through an unconstrained neural map $f_\theta$; then define the model as the finite average
$q_\theta(x) \;=\; \frac{1}{N_c}\sum_{i=1}^{N_c} q(x\mid w_{\theta,i}),\quad w_{\theta,i}=f_\theta(z_i)$.
Crucially, $w$ is resampled for every density evaluation and sampling call (Fig. 2; §2.2). The paper argues this yields (i) exact evaluation for the finite mixture in Eq. (2), (ii) single‑step sampling, (iii) architectural freedom (no bijectivity/Jacobians/ODEs), (iv) the option to choose $p_{\text{base}}$ in lower dimension to “learn manifolds,” and (v) efficient training with forward or reverse KL. Experiments cover 2‑D toy densities, reverse‑KL fitting without observations, conditional SBI, SPD matrices via Wishart, and latent‑space “manifolds” for MNIST/JAFFE; runtime plots (Fig. 3) emphasize speedups over NF/FM/FFF.

**Strengths:**

- Simplicity & speed: No Jacobians/ODEs; likelihood is a sum over $N_c$. Runtime plots (Fig. 3) back the speed advantage.
- Flexible component families: Wishart experiment on SPD matrices is a nice demonstration (Fig. 9).
- Both KL directions: Efficient sampling and evaluation allow reverse‑KL training without tricks; synthetic results look competitive with NFs in reverse‑KL (Fig. 8).
- Clear exposition & code: Equations (5–11) and Appendix A.1 make reproduction feasible, and specific $N_c$ choices are stated for some tasks.

**Weaknesses:**

1. **Stochastic “exact” likelihood & evaluation protocol.**.
Likelihoods depend on a fresh resample of w every call (pp. 3–4; Fig. 2). Report NLL mean+/-std over resamples, define a fixed‑mixture evaluation protocol (freeze one draw of w at test time), and separate “finite‑mixture exactness” from “marginal exactness.”
2. **No guidance on $N_c$ and no scaling study.**.
Provide NLL/accuracy vs $N_c$ curves, wall‑clock trade‑offs, and dimension sweeps. Connect observations to known nonparametric rates to discuss when growth of $N_c$ becomes prohibitive.
3. **Manifold claims need precision or a different construction.**.
Either adopt degenerate/anisotropic components (with care for normalization) or manifold‑native densities to achieve genuine lower‑dimensional support; otherwise rephrase as mass concentrated near low‑dimensional structure.
4. **Novelty vs MDN not delineated.**.
Explicitly compare (conceptually and experimentally) to Mixture Density Networks [MDN] (network‑parameterized mixtures/kernels). Discuss statistical/compute advantages beyond not needing to evaluate $q_\theta(w)$. Does this limit novelty? Also, can you show that the approach is going to learn something different than a Kernel Density Estimation method?
5. **Comparisons emphasize low-dimensional micro‑benchmarks.**.
Mixture of Gaussians does not scale well with ambient space dimension $D$; curse of dimensionality. Theoretical properties of $N_c$ as a function of $D$ are missing. Introduce higher dimensional experiments to convince otherwise, e.g. on typical image generative modelling datasets (MNIST; CelebA; ImageNet 64x64).
6. **Universal approximation theory is missing.**.
Even light‑touch results would help: e.g., approximation error of the Monte‑Carlo mixture to the ideal marginal as a function of $N_c$; and conditions under which the model inherits known universal approximation properties of mixtures (with references).

References
----
[MDN] Mixture Density Networks. Bishop 1994.

**Questions:**

1. Test‑time protocol & variance. Do you freeze one draw of $\{w_{\theta,i}\}_{i=1}^{N_c}$ for the entire test set, or resample per‑example? Please report NLL mean+/-std over 20 resamples on your test sets and clarify which numbers appear in figures. The paper currently implies resampling at every evaluation (pp. 3–4; Fig. 2).
2. Guidance on N_c. Provide ablations of NLL/C2ST vs $N_c$ and discuss compute/variance trade‑offs. In Appendix A.4 you pick $N_c=$ half the training set, SBI uses $N_c=2048$, JAFFE $N_c=128$; what drives these choices? For latent-variable MNIST, is the VAE trained jointly with the Marginal Flow? What is the training procedure exactly?
3. Dimension scaling. Have you tried ambient $D \ge 32$ in data space (not latent) with quantitative benchmarks? How does runtime and NLL change with d when keeping a fixed compute budget? Report test log-likelihood and sample quality metrics (FID,KID) comparing against other generative modelling methods using networks with similar sizes.

---

> ### Author Response · Authors · 2025-11-22
>
> We would like to thank the reviewer for their detailed feedback and for highlighting the importance of analyzing the scaling behaviour with $N_c$. We also thank the reviewer for acknowledging the simplicity, efficiency and flexibility of the proposed approach.
>
> ## Weaknesses
> > 1. Stochastic “exact” likelihood & evaluation protocol. Likelihoods depend on a fresh resample of w every call (pp. 3–4; Fig. 2)[...]”
>
> We thank the reviewer for highlighting the need for clarity when using the term exactness for density evaluation. As detailed in our global response, our framework supports two valid interpretations (independent of training). When we say exact density we refer to interpretation (2.), i.e. the distribution defined by Eq. (2) which can be evaluated exactly given $N_c$ mixtures. In the experiments at evaluation time we consider the model as fixed and use the same set of $N_c$ mixtures for evaluating all test points. Variance in the results is reported when significant, e.g. Figure 9.
>
> > 2. No guidance on $N_c$ and no scaling study. Provide NLL/accuracy vs $N_c$ curves, wall‑clock trade‑offs, and dimension sweeps. Connect observations to known nonparametric rates to discuss when growth of $N_c$ becomes prohibitive.
>
> We thank the reviewer for the suggestion, shared with other reviewers as well. We now address this aspect rigorously by analyzing the bias and variance of the KL estimator approximated over $N$ samples as function of $N_c$, as outlined in the general response and in the Appendix A5-A7. The bias of our estimator scales as $\mathcal{O}(1/N_c)$ and the variance as $\mathcal{O}(1/N + 1/(N \cdot N_c))$. Crucially, the constant factor in these bounds depends on the intrinsic dimension of the data manifold and only linearly in the ambient dimension (for the variance). This suggests that under the manifold hypothesis, the growth of $N_c$ does not become prohibitive in high dimensions.
>
> > 3. Manifold claims need precision or a different construction. Either adopt degenerate/anisotropic components (with care for normalization) or manifold‑native densities to achieve genuine lower‑dimensional support.
>
> We agree with the reviewer that we are not learning a normalized density strictly on the manifold, but rather a density in the ambient space that concentrates mass near a lower-dimensional structure. This is by design: we assume the data is generated from a lower-dimensional manifold and observations are noisy. Our goal is to model the observed ambient distribution while also concentrating the density on a lower dimensional structure, which effectively renders the manifold hypothesis. By defining the base distribution on a lower-dimensional domain and mapping it to the ambient space via $f_\theta$, we effectively capture the manifold topology while maintaining a valid density in the ambient space (Figure 4 and Figure 9). We will update the text to clarify what we mean with manifold.
>
> > 4. Novelty vs MDN not delineated. Explicitly compare (conceptually and experimentally) to Mixture Density Networks [MDN] (network‑parameterized mixtures/kernels). Discuss statistical/compute advantages beyond not needing to evaluate $q_\theta(w)$ Does this limit novelty?
>
> We thank the reviewer for mentioning relevant related work. We would like to highlight the differences between the two models. First, MDNs are typically designed for conditional density estimation (outputting mixture parameters per input $x$), whereas Marginal Flow models the global marginal density via latent parameters $w$. Conceptually, an MDN attempts to cover the data distribution with a set of discrete mixtures. In contrast, Marginal Flow learns a continuous function $f_\theta$ that traces the data manifold. Integration over $w$ effectively allows us to model continuous topologies (like spirals) rather than discrete clusters. A visual difference can be seen in Figure 1, where we compare Marginal Flow with a mixture model where resampling of the means is disabled. Lastly, MDNs requires observations to be trained while Marginal Flow can be trained without observations on unnormalized targets.
>
> > 4. Also, can you show that the approach is going to learn something different than a Kernel Density Estimation method?
>
> The fundamental differences between KDE and Marginal Flow are two-fold: representation efficiency and structure learning. First, KDE effectively places a kernel at every data point ($N_c = N_{\text{data}}$), which is computationally expensive, while Marginal Flow resamples the mixture at every iteration. Effectively, during training far fewer mixtures are required (See Figure 1). Note that with Marginal Flow we can learn other hyperparameters (e.g. variance/bandwidth) alongside $f_\theta$ directly during training. Second, Marginal Flow learns a parametric mapping $f_\theta$ that compresses the data distribution into a lower-dimensional latent representation. This would not be possible with KDE. Both aspects are crucial in many applications

---

> > ### Author Response · Authors · 2025-11-22
> >
> > > 5. Comparisons emphasize low-dimensional micro‑benchmarks.. Mixture of Gaussians does not scale well with ambient space dimension $D$; curse of dimensionality. Theoretical properties of $N_c$ as a function of $D$ are missing. Introduce higher dimensional experiments to convince otherwise, e.g. on typical image generative modelling datasets (MNIST; CelebA; ImageNet 64x64).
> >
> > We acknowledge the concern regarding the curse of dimensionality, shared by all density estimation models. However, our method avoids this precisely by exploiting the manifold hypothesis. In the bias and variance analysis we included in the global response and in Appendix A5-A7 we show this argument rigorously. The scaling constants depend on the intrinsic dimension $m$, not the ambient dimension $d$. Furthermore, the experiments on the Wishart distributions ($d=55$ and $d=5050$) show that this is not a problem also in practice. As shown in the updated Figure 9, Marginal Flow successfully recovers the true manifold and outperforms baselines in these high-dimensional settings.
> > TODO Regarding typical image modelling datasets, we have experiments with MNIST and JAFFE which provide interesting manifolds.
> >
> > > 6. Universal approximation theory is missing.. Even light‑touch results would help: e.g., approximation error of the Monte‑Carlo mixture to the ideal marginal as a function of $N_c$; and conditions under which the model inherits known universal approximation properties of mixtures (with references).
> >
> > We thank the reviewer for highlighting the importance of a rigorous treatment of the universality, hence the connection between Eq 1 and Eq 2. Marginal Flow Eq. (2) is an unbiased estimator of the marginal distribution Eq. (1). The bias comes into play when using objective functions like the KL divergence, which requires log-probabilities. Any mixture models will then suffer from the bias introduced by Jensen's inequality. We have now analysed the resulting bias and variance incorporating existing theory on nested Monte Carlo estimators. We showed that the bias decays as $1/N_c$. Crucially, the constants depend on the manifold dimension $m$ (base distribution dimensionality) and is independent of the ambient dimension $d$. For a more detailed discussion we refer to the global response.
> >
> > ## Questions
> >
> > > 1. Test‑time protocol & variance. Do you freeze one draw of $\{w_{\theta,i}\}_{i=1}^{N_c}$ for the entire test set, or resample per‑example? Please report NLL mean+/-std over 20 resamples on your test sets and clarify which numbers appear in figures. The paper currently implies resampling at every evaluation (pp. 3–4; Fig. 2).
> >
> > In our experiments, we utilized the interpretation (2) (see Global response). At test time, we computed the log densities over a fixed set of (seeded) $N_c$ mixtures. When we observed variance in the estimators we reported variance (newly also in the previously missing Figure 9). This aligns with our theoretical variance analysis.
> >
> > > 2. Guidance on N_c. Provide ablations of NLL/C2ST vs $N_c$ and discuss compute/variance trade‑offs. In Appendix A.4 you pick $N_c=$ half the training set, SBI uses $N_c=2048$, JAFFE $N_c=128$; what drives these choices? For latent-variable MNIST, is the VAE trained jointly with the Marginal Flow? What is the training procedure exactly?
> >
> > Thanks to the additional analysis in Appendix A5-A7 on the bias and variance term it is now possible to make an informative choice about $N$ and $N_c$ (given a fixed budget). For some guidance we refer to [1] and [2] , which already studied the problem. During training the resampling allows to use fewer $N_c$. Our choices for $N_c$ are heuristic, but our new analysis allows for optimal budget allocation between batch size $N$ and components $N_c$. SBI required more mixtures because we learned density conditioned on many variables. Since it didn't consistute a computational botleneck in terms of runtime (Figure 3), we simply picked $N_c$ high enough to make sure it would work on all settings.
> >
> > Regarding MNIST, the VAE is pre-trained to obtain the latent space, and Marginal Flow is trained subsequently to model the density of these fixed latents.
> >
> > > 3 Dimension scaling. Have you tried ambient $D>32$ in data space (not latent) with quantitative benchmarks?
> >
> > Yes, we have explicitly tested this scenario in the two experiments with Wishart distributions, where the ambient dimension is $D=55$ and $D=5050$. Marginal Flow was able to accurately model the distribution and identify the manifold structure. In terms of runtime Marginal Flow is very efficient (Figure 3) while for NLL we refer to the analysis in Appendix A5-A7, which shows that it is governed by the intrinsic dimensionality $m$ and not by the ambient dimensionality $d$
> >
> > [1] Hong and Juneja, Estimating the mean of a non-linear function of conditional expectation, WSC, 2009.
> >
> > [2] Rainforth et al., On nesting monte carlo estimators, ICML, 2018.

---

### Author Response · Authors · 2025-11-21

## Global response
We thank reviewers for their time and constructive feedback. We were encouraged that reviewers found the method **simple, fast, and flexible**. We have addressed the main concerns regarding **scaling behavior**, **exactness definition**, and **distinction from VAEs** by adding a rigorous theoretical analysis of the estimator's bias/variance and clarifying the model definition.

### 1. Theoretical scalability and high-dimensional performance
A primary concern was the scaling behavior of the integral approximation based on the number of components $N_c$ and the performance in high dimensions. We have added a rigorous theoretical analysis (Appendix A.5–A.7), leveraging existing theory on nested Monte Carlo estimators [1, 2], to derive explicit bounds for the bias and variance of a finite approximation $\widehat{\mathcal{L}}(N,N_c)$ of the KL divergence evaluated over $N$ samples: $\mathrm{Bias}(\widehat{\mathcal{L}}(N,N_c)) \approx - \frac{\gamma}{2 N_c}$ and $\mathrm{Var}(\widehat{\mathcal{L}}(N,N_c)) \approx \frac{V_{\text{data}}}{N} + \frac{\gamma}{N N_c}$

* **Theoretical Bounds:** These bounds demonstrate that the bias scales as $\mathcal{O}(1/N_c)$ and variance as $\mathcal{O}(1/N) + \mathcal{O}(1/(N \cdot N_c))$. Crucially, the constant $\gamma$ depends on the **intrinsic manifold dimension** $m$, not the ambient dimension $d$. Under the manifold hypothesis ($m \ll d$), Marginal Flow avoids the curse of dimensionality by exploiting this lower-dimensional structure. Following [1, 2], we can also optimally allocate the computational budget between $N$ and $N_c$.
* **High-Dimensional Validation:** Our experiments confirm this scaling behavior in high-dimensional settings. We trained Marginal Flow to learn mixtures of Wishart distributions via reverse KL divergence with **$d=55$** ($10 \times 10$ matrices) and in the very high-dimensional setting of **$d=5050$** ($100 \times 100$ matrices) via log-likelihood. Marginal Flow outperformed Normalizing Flows in both cases while accurately recovering the true manifold.

### 2. Model definition: exactness and interpretations
Reviewers asked for clarification concerning the term *exactness*. We believe, the reviewers had a different interpretation than the authors. Both interpretations are valid but lead to a different conclusion on exactness. We first outline the reviewer's and then our's:
1. **Monte Carlo estimate of marginal:** Interpretation: Eq. 1 describes the modelled distribution. The model density $q_\theta(x)$ (Eq. 2) is an unbiased Monte Carlo estimator of the true marginal distribution (Eq. 1). While the log-likelihood loss is biased due to the nonlinearity of the log function, we now explicitly quantify and control this bias via $N_c$. With the derived expressions for bias and variance, the scaling of the approximation with $N_c$ is fully characterized.
2. **Distribution over mixtures:** Interpretation: Eq. 2 describes the modelled distribution. Conditioned on a *fixed* set of sampled parameters $w_i$, the density of the finite mixture (Eq. 2) is evaluated **exactly**. This interpretation allows for deterministic evaluation at test time (by fixing the seed). Furthermore, the randomness arising from mixture sampling can be treated as a feature rather than a bug,
providing a distribution over probabilities that captures uncertainty, which is valuable in scientific applications.

### 3. Comparison with VAEs
We clarify the fundamental differences between Marginal Flow and VAEs:
* **Posterior vs. Marginal:** VAEs are designed to approximate the **posterior** (using the marginal likelihood only to derive the ELBO). In contrast, Marginal Flow is not Bayesian; it has no notion of posterior or prior over data instances. It is designed to model the **marginal distribution** directly.
* **Global vs. Local Latents:** VAEs marginalize over *local* latents $z$ specific to a datapoint $x$ (requiring an encoder $q(z|x)$). Marginal Flow marginalizes over *global* parameter latents $w$ that define the density shape itself. We integrate over the space of possible densities, not data representations.
* **Training Flexibility:** Unlike VAEs, which require data samples $x$ to be mapped into latents, Marginal Flow models the density by construction. This allows training via Reverse KL divergence on unnormalized targets (e.g., for Simulation-Based Inference), a capability standard VAEs lack.
* **Objective Function:** Unlike VAEs, which must optimize a lower bound (ELBO) because the marginal is intractable, Marginal Flow optimizes the finite mixture log-likelihood directly. Computation of the marginal likelihood is possible also with VAEs but is more expensive and suffers from the well-known poor posterior approximation.

[1] Hong and Juneja, *Estimating the mean of a non-linear function of conditional expectation*, WSC, 2009.

[2] Rainforth et al., *On nesting monte carlo estimators*, ICML, 2018.

---

> ### Comment · Reviewer_R9aj · 2025-11-22
> **Response to general rebuttal**
>
> 1. I would still like to see experiments on typical image generative modelling datasets (MNIST; CelebA; ImageNet 64x64), without resulting to prior dimensionality reduction by another generative model. If the intrinsic dimension is small enough, then yes the model may be able to model it with sufficient $N_c$; but how much? Lack of experiments and theory is concerning here. The relationship of the dimensionality (ambient or "intrinsic) to $N_c$ should be clarified, since the model might still suffer from a curse of "intrinsic" dimensionality, which depending on the data to be modelled might not be small enough, as the authors hope for.
> 2. Given the authors clarification, the proposal looks extremely similar to Mixture Density Networks as I mentioned in the original review.

---

> > ### Author Response · Authors · 2025-11-24
> >
> > We would like to thank the reviewer for the prompt response and for continuing the discussion. We appreciate it a lot. If possible, I would suggest to continue the discussion in the reviewer-specific section rather than in the general rebuttal; so that we can keep topics more clearly organized, also for the AC.
> >
> > > I would still like to see experiments on typical image generative modelling datasets (MNIST; CelebA; ImageNet 64x64), without resulting to prior dimensionality reduction by another generative model.
> >
> > Most modern approaches for image generation and manipulation rely on highly crafted latent spaces. As the literature for image generations has been improved for decades it would be unfair to provide a comparison with our model. As we showed in the extensive experiment section, the primary goal of Marginal Flow is not computer vision. Nevertheless, we included two computer vision applications to show that Marginal Flow is a valuable tool for conditional image generation in latent spaces (which is a more modern approach).
> >
> > If the concern is primarily over high-dimensional settings, we refer the reviewer to the experiment on the Wishart distribution, which have $d=55$ and $d=5050$. Please note that this is a much higher space compared to MNIST (784) and ImageNet (4096). We will clarify in the paper that Marginal Flow is not designed for image generation directly in pixel space. In contrast we show applications in:
> > - synthetic datasets: training by log-likelihood and training by reverse KL (which most models cannot handle)
> > - simulation based inference
> > - distributions beyond vectors: Wishrat mixtures both by log-likelihood and reverse KL divergence
> > - distribution and automatic (conditional) manifold learning: Wishart mixtures, MNIST in latent space and JAFFE in latent space
> >
> > > If the intrinsic dimension is small enough, then yes the model may be able to model it with sufficient $N_c$; but how much? Lack of experiments and theory is concerning here.
> >
> > We refer the reviewer to Appendix A.5-A.7. The theory provides clear convergences for the KL approximation with respect to $N_c$ both for the bias and the variance. We further explicitly compute the dependency on the ambient dimensionality $d$ and base distribution dimensionality $m$. We do not believe it is possible to provide more general statements, and this problem is of course shared by other models like Normalizing Flows (and by deep learning in general). However, the flexibility of Marginal Flow allows for tackling specific settings better than other models, e.g. by tailoring $q(x|w)$ to the target distribution or by freely implementing $f_\theta$ (which does not have to be a bijection).
> >
> >
> > > The relationship of the dimensionality (ambient or "intrinsic) to $N_c$ should be clarified, since the model might still suffer from a curse of "intrinsic" dimensionality, which depending on the data to be modelled might not be small enough, as the authors hope for.
> >
> > In contrast to other approaches like Normalizing Flows, we can address the curse of dimensionality by choosing a base distribution with dimension $m \ll d$. This choice is justified in many real applications based on the manifold hypothesis, without which deep learning in high-dimensions would not be possible. For a detailed statement about how the bias and variance of our estimator depend on $m$ and not on $d$, we refer to Appendix A.5-A.7
> >
> > > Given the authors clarification, the proposal looks extremely similar to Mixture Density Networks as I mentioned in the original review.
> >
> > We respectfully disagree with the reviewer on this. The differences are two-fold. First, we learn a distributions over mixtures rather than fixed mixtures. This provides significantly different learnt densities (see Figure 1). Secondly, MDNs requires observations to be trained while Marginal Flow can be trained without observations on unnormalized targets. The ability to efficiently handle completely new training modes is indeed novel.

---

### Author Response · Authors · 2025-12-03

# Short summary of the rebuttal

## Strengths and weaknesses
- Overall **strengths**: reviewers praised the simplicity and efficiency of the model, the clarity of the manuscript and highlighted that it provides advantages compared to current approaches in some settings.
- Overall **weaknesses**: we believe that the way we originally introduced our model created significant misunderstandings. In particular, all reviewers were lead to believe our model Eq. (2) is to be intended as a Monte Carlo approximation of Eq. (1). However, this is true only as a motivation. Our model is the one defined in Eq. (2) and its density can be **evaluated exactly**. Most questions from reviewers concerned the quality of the Monte Carlo approximation (under the assumption that Eq. (2) approximates Eq. (1)). To address this, we added a detailed **bias and variance analysis** (Appendix A5-A7) and showed that bias (with respect to Eq. (1)) scales as $1/N_c$ while variance scales as $1/N +1/NN_c$, hence similarly in the mixtures components $N_c$ and in the samples $N$ approximating the KL divergence. We also show that the curse of dimensionality can be avoided by choosing a base distribution with a lower dimensional support.


## Rebuttal details
- reviewer **R9aj** highlighted the simplicity, speed, flexibility of the model, and also that it allows for new efficient training objectives. The main concerns were exactness of the density evaluation, scalability and the need for theoretical insights into the scaling behavior. We addressed such concerns by specifying that our model in Eq. (2) is indeed exact, and we provided a bias and variance analysis that clarifies scalability with $N_c$. We further highlighted differences with MDN and that we already had high-dimensional experiments ($d=5050$). The reviewer asked for further experiments with images in the pixel-space. We believe that this goes beyond the scope of our model because (i) most recent approaches operate on latent spaces (as we did in two experiments) and (ii) we showcased how powerful the model is already with simple isotropic mixture components but images in pixel-space would require structured mixture components (which our framework allows but go beyond the scope of this paper).
- reviewer **nQRv** expressed three main criticisms: (i) scalability in high dimensions, (ii) behavior for $N_c\rightarrow\infty$ and (iii) comparison with VAE. (i/ii) With the variance and bias analysis it is clear that $N_c$ is not a limiting factor and that the curse of dimensionality can be avoided (e.g. Figure 9 right has $d=5050$). (iii) we included a detailed comparison to VAE in the global response. The reviewer raised the score from 0 to 2 highlighting that we should also clarify better in the paper the scope of the model and its position among latent variable models.
- reviewer **sA1N** highlighted that the method is clearly illustrated and presents advantages compared to current approaches in some settings. The reviewer expressed criticism concerning scalability in high dimensions, the need for sensitivity analysis on $N_c$ and that marginalization and Monte Carlo are not novel approaches. We showed theoretically that bias scales as $1/N_c$ while variance scales as $1/N +1/NN_c$. Furthermore, the paper originally included a high dimensional experiment in $d=5050$ (Figure 9, right). Lastly, we clarified that Eq. (1) serves only as a motivation for our model in Eq. (2), which does not aim to approximate Eq. (1) and has a density that can be easily evaluated. Compared to other (classical) mixture models, we learn the distribution to sample the mixtures from and constantly resample the mixtures during training.

We acknowledge the raised concerns (especially in the pre-rebuttal version), but continue to believe in 1) the **relevance** of our model, as we outperform current SOTA methods. 2) the **novelty**, as we learn a distribution over mixture components. 3) the **flexibility**, as we are free to choose the conditional family $q(x|w)$ depending on the application and allow for any arbitrary losses.

We would like to thank the AC for the (additional) time invested in evaluating our work.

---

### Note · Authors · 2026-01-16

**Comment:**

Dear Reviewers and AC,

We sincerely thank the Reviewers and the AC for their time and for recognizing the novelty and efficiency of our approach.

The discussion regarding "exactness" was particularly valuable; it highlighted that our current presentation caused ambiguity regarding the model’s definition, obscuring the fact that the underlying method is sound. We are grateful for this feedback, as it allows us to correct the exposition and scope of our work. Furthermore, we agree that we should highlight early on the differences with VAEs, which are significant both in terms of scope and applicability.

Best regards,
The authors

**Withdrawal Confirmation:**

I have read and agree with the venue's withdrawal policy on behalf of myself and my co-authors.